# Structural basis for the multi-activity factor Rad5 in replication stress tolerance

Miaomiao Shen [1,2,3], Nalini Dhingra[4], Quan Wang [5], Chen Cheng[6], Songbiao Zhu[7], Xiaolin Tian[7], Jun Yu[8], Xiaoxin Gong[1,2,3], Xuzhichao Li[1,2,3], Hongwei Zhang[8], Xin Xu [1,2,3], Liting Zhai[8], Min Xie[8], Ying Gao[8], Haiteng Deng[7], Yongning He[6], Hengyao Niu[5], Xiaolan Zhao [4] & Song Xiang [1,2,3 ✉]

The yeast protein Rad5 and its orthologs in other eukaryotes promote replication stress tolerance and cell survival using their multiple activities, including ubiquitin ligase, replication fork remodeling and DNA lesion targeting activities. Here, we present the crystal structure of a nearly full-length Rad5 protein. The structure shows three distinct, but well-connected, domains required for Rad5's activities. The spatial arrangement of these domains suggest that different domains can have autonomous activities but also undergo intrinsic coordination. Moreover, our structural, biochemical and cellular studies demonstrate that Rad5's HIRAN domain mediates interactions with the DNA metabolism maestro factor PCNA and contributes to its poly-ubiquitination, binds to DNA and contributes to the Rad5-catalyzed replication fork regression, defining a new type of HIRAN domains with multiple activities. Our work provides a framework to understand how Rad5 integrates its various activities in replication stress tolerance.

[1] Department of Biochemistry and Molecular Biology, Tianjin Medical University, 300070 Tianjin, P. R. China. [2] Key Laboratory of Immune Microenvironment and Disease (Ministry of Education), Tianjin Medical University, 300070 Tianjin, P. R. China. [3] The province and ministry co-sponsored collaborative innovation center for medical epigenetics, Tianjin Medical University, 300070 Tianjin, P. R. China. [4] Molecular Biology Department, Memorial Sloan Kettering Cancer Center, New York, NY 10065, USA. [5] Department of Molecular and Cellular Biochemistry, Indiana University Bloomington, Bloomington, IN 47405, USA. [6] State Key Laboratory of Molecular Biology, Shanghai Institute of Biochemistry and Cell Biology, CAS Center for Excellence in Molecular Cell Science, Chinese Academy of Sciences, 201210 Shanghai, P. R. China. [7] MOE Key Laboratory of Bioinformatics, Center for Synthetic and Systematic Biology, School of Life Sciences, Tsinghua University, 100084 Beijing, P. R. China. [8] CAS Key Laboratory of Nutrition, Metabolism and Food safety, Shanghai Institute of Nutrition and Health, Shanghai Institutes for Biological Sciences, Chinese Academy of Sciences, 200031 Shanghai, P. R. China. ✉email: xiangsong@tmu.edu.cn

Many types of genome lesions can impair the progression of DNA replication fork and hamper accurate genome duplication and cell survival. The multiple activities of the budding yeast Rad5 protein help cells to cope with these disruptive events, which greatly increase during replication stress. Rad5 possesses a ubiquitin ligase activity and catalyzes the Lys63-linked ubiquitin-chain modification of the proliferating cell nuclear antigen (PCNA). This modification can trigger the error-free branch of DNA damage tolerance (DDT), allowing DNA synthesis via template switch[1–4]. Rad5 also possesses ATP hydrolysis-driven DNA translocase activity, through which it catalyzes the regression of replication forks[5]. Replication fork regression is a universal and regulated response to replication stresses in eukaryotes, which can lead to multiple consequences such as stabilizing the stalled replication fork and allowing lesion bypass[6,7]. In addition, Rad5 recognizes stressed replication forks or DNA damage sites and recruits the translesion DNA polymerase Rev1 to these locations to promote DNA synthesis[8–10]. Orthologs of Rad5 have been identified in many eukaryotes. In line with their critical roles in replication stress tolerance, the human Rad5 orthologs, HLTF/HIP116 and SHPRH, are implicated in many types of cancer[11,12].

A hallmark of the Rad5 family of proteins is the presence of multiple activity domains. The budding yeast *S. cerevisiae* Rad5 (*Sc*Rad5), a model for studying the Rad5 family of proteins, interacts with Rev1 through its N-terminal 30 residues[8]. The remaining part of *Sc*Rad5 contains three distinct domains, including a HIP116 and Rad5 N-terminal (HIRAN) domain, a superfamily 2 (SF2) DNA translocase motor domain of the Snf2 sub-family (Snf2 domain), and a ubiquitin ligase RING domain embedded in the Snf2 domain (Fig. 1a and Supplementary Fig. 1). These distinct domains endow Rad5 with DNA damage site recognition[9], replication fork remodeling[5], and ubiquitin ligase[1] activities. The assembly of multiple activity domains in Rad5 suggests complex functional mechanisms. Due to the current lack of structural information on the full-length proteins of Rad5 or its orthologs, how they deploy distinct activities to fulfill their multi-faceted role in replication stress tolerance is poorly understood.

We present here a crystal structure of a nearly full-length Rad5 protein. The structure and structure-guided functional studies provide insights into the coordination of Rad5's activities. Our data indicate that distinct from the recently characterized HIRAN domain in HLTF[13–15], Rad5's HIRAN domain possesses multiple activities that are critical for the Rad5-mediated processes.

## Results

**The overall structure of the *K. lactis* Rad5.** We screened through a number of fungal species and were able to purify a nearly full-length fragment of Rad5 from the yeast *K. lactis* (*Kl*Rad5, containing residues 163–1114, Supplementary Fig. 2a, b) and crystallize it in the absence of DNA. *Kl*Rad5 shares 46.7% sequence identity with *Sc*Rad5. The structure was determined using mercury anomalous diffraction signals from mercury-derivatized crystals. Structures of the mercury-derivatized and native crystals were refined to resolutions of 3.3 Å and 3.6 Å, respectively (Table 1). Minimal differences were observed between these structures. In the remainder of the paper, we discuss the structure of the mercury-derivatized crystal because of its higher resolution.

The structure revealed an elongated shape of *Kl*Rad5 spanning 140 Å in the longest dimension (Fig. 1b). The monomeric crystal structure docks well into the density determined by negative staining electron microscopy (EM, Supplementary Fig. 3a–c) and is consistent with a molecular weight of 120 kDa determined by dynamic light scattering, suggesting that *Kl*Rad5 adopts a monomeric structure in a solution similar to the crystal and EM structures. The HIRAN, Snf2, and RING domains are well resolved in the structure. Each individual domain generally resembles previously reported homologous structures (Supplementary Fig. 4a–d), but with important differences (detailed below). The HIRAN domain and the two lobes in the Snf2 domain are nearly aligned in a line, the RING domain is located on the side. Extensive interactions are observed between the Snf2 and HIRAN domains (Fig. 1c), and between the two lobes of the Snf2 domain (Fig. 2a). In contrast, the RING domain has little contact with either the Snf2 or the HIRAN domain. Its active site for ubiquitin transfer superimposes well with equivalent regions in reported RING domain structures[16] (Supplementary Fig. 4d–e) and is fully solvent-exposed (Supplementary Fig. 4f).

**Structural basis for the Snf2 domain activation by DNA.** SF2 enzymes hydrolyze ATP to drive nucleic acid translocation. They contain seven conserved motifs distributed in two lobes.

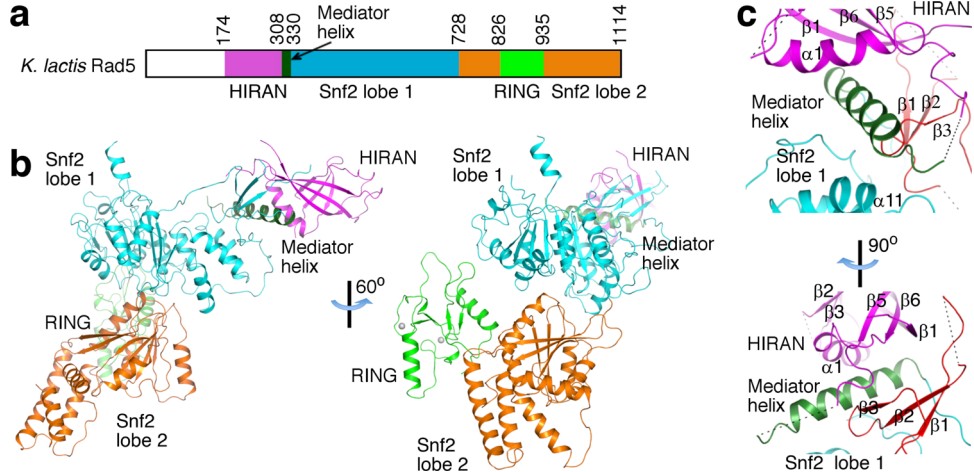

**Fig. 1 Structure of *Kl*Rad5. a** Domain organization of Rad5. The HIRAN and RING domains, the Snf2 domain lobes 1 and 2, and the mediator helix are indicated with different colors. The coloring scheme is used throughout the manuscript unless otherwise indicated. Domain boundaries for the *K. latics* Rad5 are indicated. **b** Crystal structure of *Kl*Rad5. Structural figures were prepared with PyMOL (http://pymol.org). **c** The mediator helix mediates interactions between the Snf2 and HIRAN domains. The β1–β3 insertion in the Snf2 domain lobe 1 is highlighted in red. Dashed lines indicate disordered regions in our structure.

**Table 1 Data collection and structure refinement statistics.**

|  | Native crystal | Mercury-derivatized crystal |
|---|---|---|
| *Data collection* | | |
| Space group | P6$_2$22 | P6$_2$22 |
| *Cell dimensions* | | |
| a, b, c (Å) | 188.54, 188.54, 198.23 | 187.63, 187.63, 197.54 |
| α, β, γ (°) | 90, 90, 120 | 90, 90, 120 |
| Resolution (Å) | 50.0–3.60 (3.66–3.60)* | 50.0–3.30 (3.36–3.30) |
| R$_{merge}$ | 0.076 (1.406) | 0.132 (3.566) |
| I/σI | 30.89 (1.53) | 39.75 (1.80) |
| CC$_{1/2}$ | 0.997 (0.524) | 1.001 (0.548) |
| Completeness (%) | 99.8 (99.0) | 100.0 (100.0) |
| Redundancy | 6.9 (7.2) | 56.1 (50.7) |
| *Refinement* | | |
| Resolution (Å) | 50.0–3.60 (3.73–3.60) | 50–3.30 (3.35–3.30) |
| No. of reflections | 24,440 (2247) | 31,265 (1993) |
| R$_{work}$/R$_{free}$ | 0.204/0.239 (0.304/0.338) | 0.191/0.229 (0.350/0.370) |
| *No. of atoms* | | |
| Protein | 6566 | 6791 |
| Ligand/ion | 2 | 11 |
| *B-factors* | | |
| Protein | 182.6 | 124.7 |
| Ligand/ion | 100.9 | 138.2 |
| *R.m.s. deviations* | | |
| Bond lengths (Å) | 0.002 | 0.002 |
| Bond angles (°) | 0.406 | 0.496 |

One native crystal was used for diffraction data collection, data from three mercury-derivatized crystals were merged and presented.
*Values in parentheses are for the highest-resolution shell.

DNA-binding cleft (Fig. 2c) and found that they caused strong reductions in *Kl*Rad5's affinity to dsDNA (Fig. 2d) and its dsDNA-stimulated ATPase activity (Fig. 2e). The substituted proteins behave similarly as the wild-type *Kl*Rad5 on a size-exclusion column (Supplementary Fig. 2a), suggesting that the substitutions did not affect the overall protein structure. Fourth, we introduced the E628A and Q1051D substitutions in *Kl*Rad5 that are expected to disrupt the binding of the attacking water molecule for ATP hydrolysis[17] (Supplementary Fig. 5e), and found that they did not affect the overall protein folding (Supplementary Fig. 2a) but severely inhibited the dsDNA-stimulated ATPase activity (Fig. 2e). Fifth, the ATPase-inactive conformation observed in our structure is stabilized by a hydrogen bond between Ser553 in the Snf2 domain lobe 1 and the Glu953 side chain in lobe 2 (Fig. 2b), which is not expected to interact with lobe 1 in the ATPase-active form (Supplementary Fig. 5c). We removed this hydrogen bond by the E953A substitution and found that it did not affect the overall protein folding (Supplementary Fig. 2a) but increased the Vmax of *Kl*Rad5's dsDNA-stimulated ATPase activity by 30% (Fig. 2e). Finally, we performed hydrogen-deuterium exchange experiments (Supplementary Fig. 6a–b) and found that dsDNA reduced deuterium uptake of peptides 548–554, 684–700, 1039–1049, and 1081–1092 in *Kl*Rad5, which are expected to lose solvent accessibility upon dsDNA binding. The Snf2 domain lobes 1 and 2 are modeled as rigid bodies in our dsDNA-bound model of *Kl*Rad5, in which helices α2 and α8 in lobe 2 clashes with lobe 1 and DNA (Supplementary Fig. 6a). We found that peptides 771–778 and 1093–1108 in these helices have increased deuterium uptake upon dsDNA binding, suggestive of local conformational change to resolve the clash. Collectively, these data support our hypothesis that *Kl*Rad5's Snf2 domain rests in an ATPase-inactive conformation and undergoes a large conformational change upon binding dsDNA to an ATPase-competent state.

**The HIRAN domain is critical for the Rad5-catalyzed PCNA-anchored ubiquitin-chain extension.** Rad5 catalyzes the ubiquitin-chain modification of PCNA that triggers the error-free DDT[1,2,4,11]. A yeast two-hybrid study suggested that the HIRAN domain in *Sc*Rad5 mediates interaction with PCNA[9]. Biochemical studies indicated that *Sc*Rad5 interacts with PCNA, catalyzes PCNA-anchored ubiquitin-chain extension in vitro, and its N-terminal 500 residues encompassing the HIRAN domain is required for efficient PCNA ubiquitination[27,28]. We found that *Kl*Rad5 also possesses PCNA-binding and PCNA-anchored ubiquitin-chain extension activities, both activities were strongly inhibited by truncating its HIRAN domain (ΔHIRAN) (Fig. 3a, b and Supplementary Fig. 7a). In contrast, the ΔHIRAN truncation did not affect *Kl*Rad5's activity in stimulating unanchored ubiquitin-chain extension by Ubc13-Mms2[27] (Fig. 3c and Supplementary Fig. 7b). As a control, the R906E substitution disrupting the RING-ubiquitin interaction mediated by the highly conserved Arg906[16] in *Kl*Rad5 severely inhibited both PCNA-anchored and unanchored ubiquitin-chain extension (Fig. 3b, c and Supplementary Fig. 7a, b). Neither the ΔHIRAN truncation nor the R906E substitution affected the overall protein folding of *Kl*Rad5 (Supplementary Fig. 2a). Together, these data suggest that *Kl*Rad5's HIRAN domain makes a critical contribution to PCNA binding and recruits it for poly-ubiquitination, but does not affect *Kl*Rad5's ubiquitin ligase activity per se.

PCNA forms a trimer and has a predominantly negatively charged outer surface (Supplementary Fig. 7c)[29]. *Kl*Rad5's HIRAN domain contains a positively charged region (Fig. 2c and Supplementary Fig. 5b) that is highly conserved (Fig. 3d). To

Structural studies on SF2 enzymes Vasa and NS3 revealed that motifs I and II in lobe 1 and motif VI in lobe 2 make critical contributions to ATP binding and hydrolysis[17,18]. In our structure, motifs I/II have located over 20 Å away from motif VI (Fig. 2b), suggesting that *Kl*Rad5 rests in an ATPase-inactive state in the absence of DNA. The observed Snf2 domain conformation is stabilized by extensive interactions between its lobes 1 and 2 that bury 1300 Å$^2$ of surface area (Fig. 2a). Structures of several other Snf2 family enzymes have revealed that their Snf2 domains also rest in ATPase-inactive conformations stabilized by lobe 1–2 interactions[19–21]. However, the interaction between lobes 1 and 2 and their orientation are drastically different in these structures (Supplementary Fig. 5a). Studies on Snf2 family enzymes indicated that upon binding double-strand (ds) DNA they adopt conformations similar to the ATPase-active conformation of Vasa and NS3[22–26]. A model of *Kl*Rad5 in such a conformation shows that several positively charged regions on its surface (Supplementary Fig. 5b) concentrate at the expected dsDNA-binding cleft (Fig. 2c), which could mediate favorable interactions with the negatively charged DNA backbone. These observations suggest that *Kl*Rad5's Snf2 domain can bind dsDNA, which triggers a major conformational change to form the ATPase-active site between its lobes 1 and 2 (Supplementary Fig. 5c).

We performed a number of experiments to test the above hypothesis. First, we carried out fluorescence polarization (FP, Fig. 2d) and co-precipitation (Supplementary Fig. 5d) experiments and found that *Kl*Rad5 binds dsDNA with high affinity. Second, we performed ATPase activity experiments and found that this activity of *Kl*Rad5 was dsDNA-dependent (Fig. 2e) like reported for *Sc*Rad5[5]. Third, we introduced charge-reversal substitutions R610E, R1000E, and K1023E into the predicted

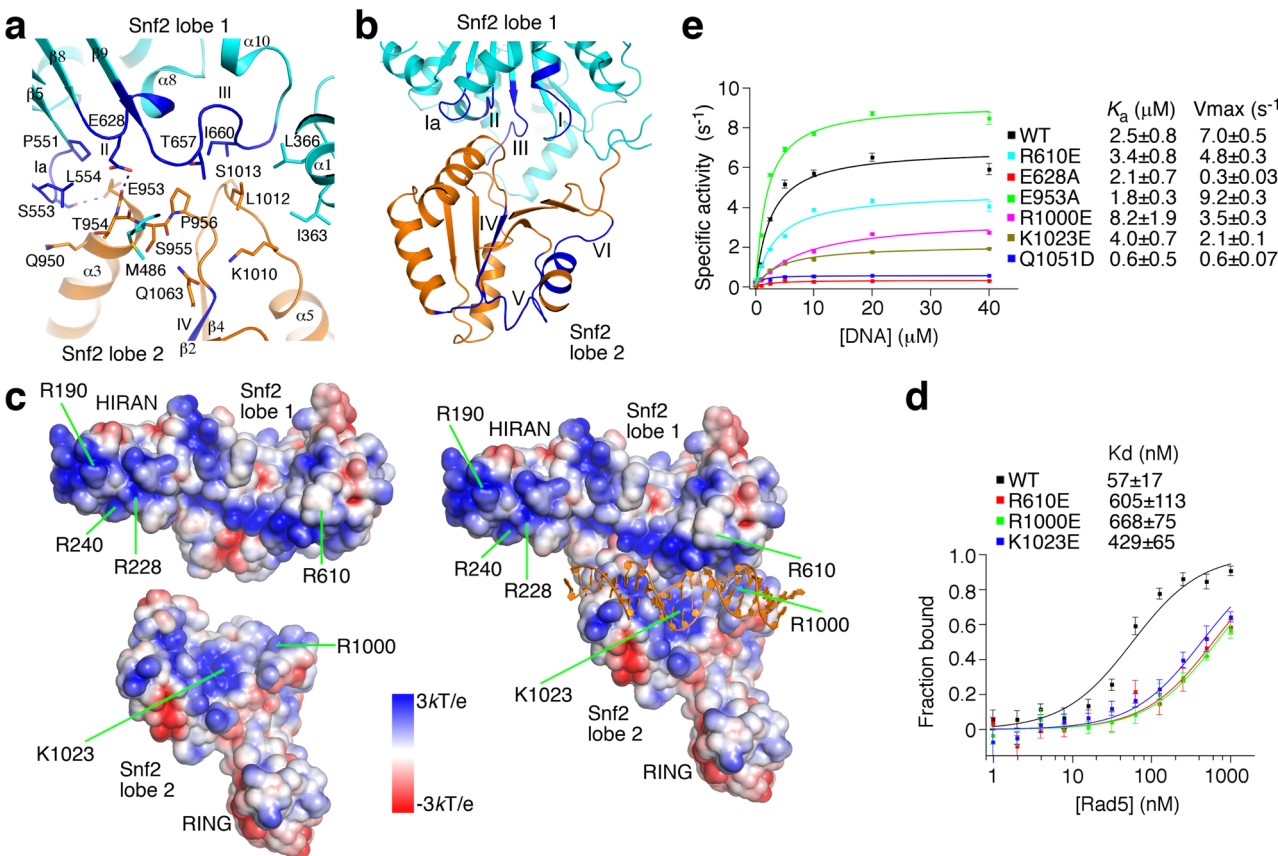

**Fig. 2 *Kl*Rad5's Snf2 domain rests in an ATPase-inactive conformation and is activated by dsDNA. a** The interface between the Snf2 domain lobes 1 and 2 observed in our structure. Residues important for the lobes 1–2 interaction are highlighted. Dashed lines indicate hydrogen bonds. **b** Locations of the conserved motifs in the Snf2 domain in our structure (blue). **c** Electrostatic potential on the *Kl*Rad5 protein surface. In the left panel, the N- and C-terminal halves of *Kl*Rad5 are shown separately. The N-terminal half contains the HIRAN domain, the mediator helix, and the Snf2 domain 1; the C-terminal half contains the Snf2 domain lobe 2 and the RING domain. In the right panel, *Kl*Rad5 is modeled in the dsDNA (orange)-bound conformation based on the Snf2-nucleosome complex structure[22]. Its N- and C-terminal halves are treated as rigid bodies in the model. **d** FP experiments probing dsDNA binding to the wild-type (WT) and substituted *Kl*Rad5. **e** dsDNA-stimulated ATPase activity of the wild-type and substituted *Kl*Rad5. Data in panels d and e are presented as mean values +/− standard deviations of three independent experiments. Errors in $K_d$, $K_a$, and $V_{max}$ are derived from data-fitting. Source data for panels d and e are provided as Source Data file.

test if this region mediates electrostatic interactions with PCNA's outer surface, we introduced charge-reversal substitutions at three conserved arginine residues in this region (R190E/R228E/R240E, 3RE), which did not affect the overall protein folding (Supplementary Fig. 2b). We found that the 3RE substitution replicated the effects of the ΔHIRAN truncation on PCNA binding (Fig. 3a), PCNA-anchored and unanchored ubiquitin-chain extension (Fig. 3b, c and Supplementary Fig. 7a, b). To further probe the Rad5-PCNA interaction we also introduced charge-reversal substitutions in the negatively charged regions in the PCNA outer surface (Supplementary Fig. 7c). We found that the D109K and E113K substitutions at the cleft between PCNA monomers severely inhibited the interaction between *Kl*Rad5 and PCNA, but substitutions at other locations in PCNA had little effect (Supplementary Fig. 7c, d). The D109K and E113K substitutions did not change the elusion profile of PCNA on a size-exclusion column (Supplementary Fig. 7e), suggesting that they did not change its overall structure. Together, these data suggest that electrostatic interactions between the positively charged region in *Kl*Rad5's HIRAN domain and the cleft between monomers in the PCNA trimer are critical for their association.

In contrast to *Kl*Rad5, it was reported that HLTF does not possess a detectable affinity to PCNA despite a putative AlkB homolog 2 PCNA-interacting motif (APIM) in its sequence[30],

and its HIRAN domain is dispensable for PCNA poly-ubiquitination[13]. The putative APIM motif is conserved between HLTF and Rad5 and our structure indicates that this region in *Kl*Rad5 (residues 1069–1073, Supplementary Fig. 1) is largely buried in the Snf2 domain lobe 2 interior and unlikely to contribute to protein–protein interaction. Consistent with the previous reports, our co-precipitation experiments did not reveal detectable interactions between PCNA and HLTF or its HIRAN domain (*Hs*HIRAN, Supplementary Fig. 7f). The HIRAN domains in HLTF and *Kl*Rad5 share 12% sequence identity. To test if the HIRAN domain in other Rad5 orthologs also binds PCNA and assists its ubiquitination, we introduced the 3RE-equivalent substitution into *Sc*Rad5 (R187E/R229E/R241E, *Sc*3RE), whose HIRAN domain shares 40% sequence identity with *Kl*Rad5's HIRAN domain. Our yeast two-hybrid experiments indicated that this substitution abolished *Sc*Rad5's interaction with PCNA (Fig. 3e). In vivo PCNA poly-ubiquitination experiments indicated that it strongly reduced the previously reported PCNA poly-ubiquitination in cells treated with the replication stress-causing agent methyl methanesulfonate (MMS)[1] (Fig. 3f). Although the *Sc*3RE substitution moderately reduced the protein level (Fig. 3g), it is unlikely that the moderate reduction alone can account for the strongly inhibited PCNA ubiquitination and our data suggest that this

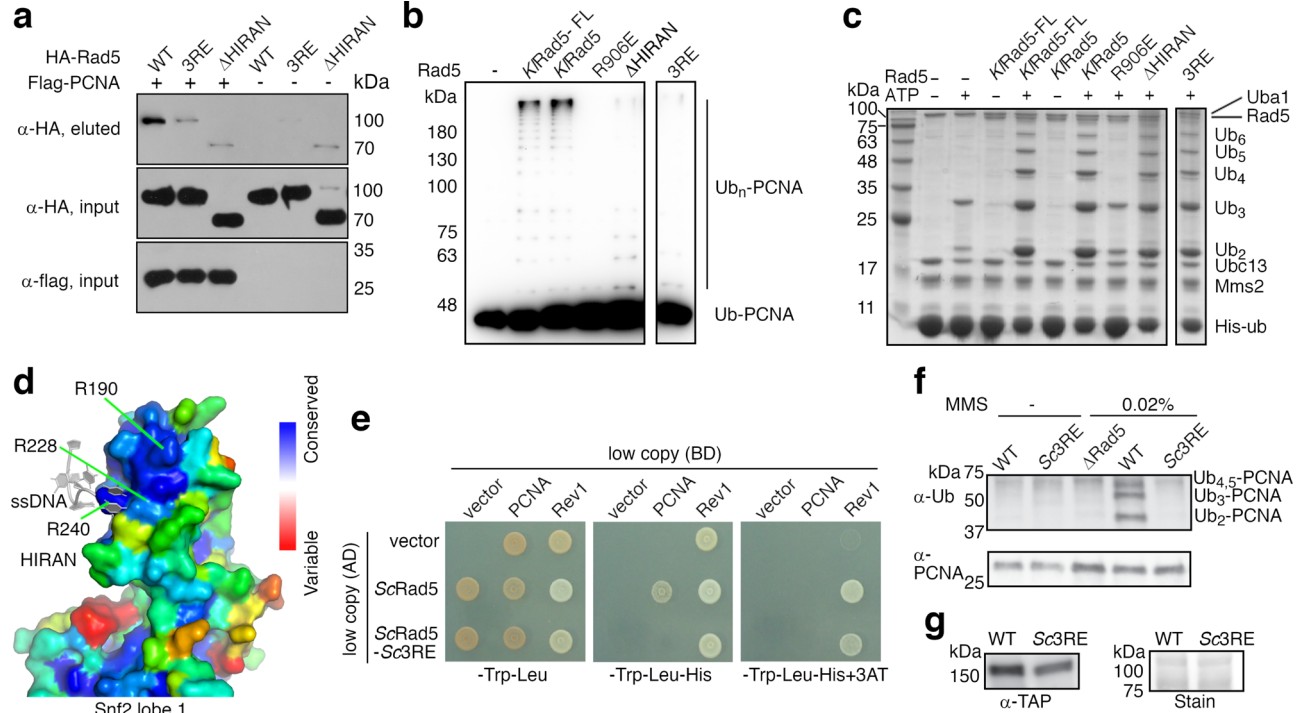

**Fig. 3 The conserved and positively charged region in Rad5's HIRAN domain interacts with PCNA and is required for its poly-ubiquitination. a** PCNA-*Kl*Rad5 co-precipitation experiments. Hemagglutinin (HA)-tagged *Kl*Rad5 co-precipitated with flag-tagged PCNA is analyzed with western blot. Three repeats of the experiment were performed, which gave similar results. **b** The 3RE substitution and the ΔHIRAN truncation severely inhibit *Kl*Rad5-catalyzed PCNA-anchored ubiquitin-chain extension. Western blot analysis against the ubiquitin (Ub)-PCNA fusion protein substrate is presented. *Kl*Rad5-FL, full-length *K. lactis* Rad5. Three repeats of the experiment were performed, which gave similar results. **c** Neither the 3RE substitution nor the ΔHIRAN truncation inhibits *Kl*Rad5-stimulated unanchored ubiquitin-chain extension by the Ubc13-Mms2 complex. Sodium dodecyl sulfate-polyacrylamide gel electrophoresis analysis of the reactions is shown. The proteins were detected with Coomassie blue staining. $Ub_2$-$Ub_6$ indicates ubiquitin-chains with 2–6 ubiquitin moieties. Uba1, the *S. cerevisiae* ubiquitin-activating enzyme. Two repeats of the experiment were performed, which gave similar results. **d** Sequence conservation of individual residues in *Kl*Rad5's HIRAN domain. The conservation is calculated based on sequence alignment of *Kl*Rad5 and 145 Rad5 homologs identified by CONSURF (http://consurf.tau.ac.il) and mapped onto the *Kl*Rad5 structure. ssDNA (gray) bound to HLTF's HIRAN domain[15] is shown for reference. **e** The *Sc*3RE substitution inhibits *Sc*Rad5's interaction with PCNA in yeast two-hybrid assay. Rev1 is included as a control. BD, Gal4 DNA-binding-domain plasmids; AD, Gal4 activation-domain plasmids; 3AT, 3-amino-l,2,4-triazole. **f** The *Sc*3RE substitution abolishes MMS-induced PCNA poly-ubiquitination in yeast. PCNA and ubiquitinated PCNA are detected with western blot. **g** The *Sc*3RE substitution moderately reduces the *Sc*Rad5 protein level. The *Sc*Rad5 protein level is probed by western blot against the tandem affinity purification (TAP) tag attached to it. Equal loading was confirmed by Ponceau S staining (right panel). At least three independent strains per genotype were characterized for experiments presented in panels **f** and **g**, which gave similar results. Source data for panels **a–c** and **f**, **g** are provided as Source Data file.

substitution strongly inhibits the *Sc*Rad5-catalyzed PCNA ubiquitination. Together, these data suggest that PCNA binding and aiding its poly-ubiquitination is a conserved function of HIRAN domains in some but not all Rad5 family members.

**Rad5's HIRAN domain contributes to DNA binding.** HLTF's HIRAN domain binds single strand (ss) DNA[13–15]. Our co-precipitation experiments revealed a strong co-precipitation of *Kl*Rad5 with ssDNA that can be suppressed by the *E. coli* ssDNA-binding protein (SSB), suggesting that *Kl*Rad5 also binds directly to ssDNA (Supplementary Fig. 8a). Strong *Kl*Rad5-ssDNA interaction was also observed in our FP experiments (Fig. 4a), which indicated that the binding is decreased by the ΔHIRAN truncation, suggesting that ssDNA binding is a common feature for the HIRAN domain in Rad5 family members. Structural alignment indicates that the ssDNA-binding site in HLTF's HIRAN domain overlaps with the conserved and positively charged region in *Kl*Rad5's HIRAN domain (Fig. 3d). We found that the 3RE substitution strongly inhibited ssDNA binding to *Kl*Rad5 (Fig. 4a), which suggests that the positively charged region in *Kl*Rad5's HIRAN domain makes a critical contribution to ssDNA binding.

Despite a shared ability in ssDNA binding, the HIRAN domains in *Kl*Rad5 and HLTF show significant differences. The β2-β3 loop that makes a significant contribution to ssDNA binding in HLTF[13–15] has a different length and amino acid composition in *Kl*Rad5 (Supplementary Fig. 1) and is largely disordered in our structure. In addition, only two of the ten residues critical for ssDNA binding in HLTF are conserved in *Kl*Rad5 (Supplementary Fig. 1). It has been reported that HLTF's HIRAN domain specifically recognizes the ssDNA 3'-hydroxyl[13–15]. Our co-precipitation experiments confirmed that the HLTF–ssDNA interaction requires the free ssDNA 3'-hydroxyl group at 100 mM salt concentration (Supplementary Fig. 8a)[15]. At lower salt concentration, HLTF binds to ssDNA with blocked 3'- or 5'-ends, suggesting that other regions in ssDNA also contribute to HLTF binding at lower salt concentration. In contrast to HLTF, we found that *Kl*Rad5 and the closely related *Sc*Rad5 bind to ssDNA with blocked 3'- or 5'-ends with a similar affinity at all the salt concentrations we tested (Supplementary Fig. 8a). Consistently, our FP experiments performed at 70 mM salt concentration did not reveal significant differences between *Kl*Rad5's affinity to ssDNA with blocked 3'- and 5'-ends (Fig. 4b). To directly test if Rad5's HIRAN domain specifically interacts with the ssDNA

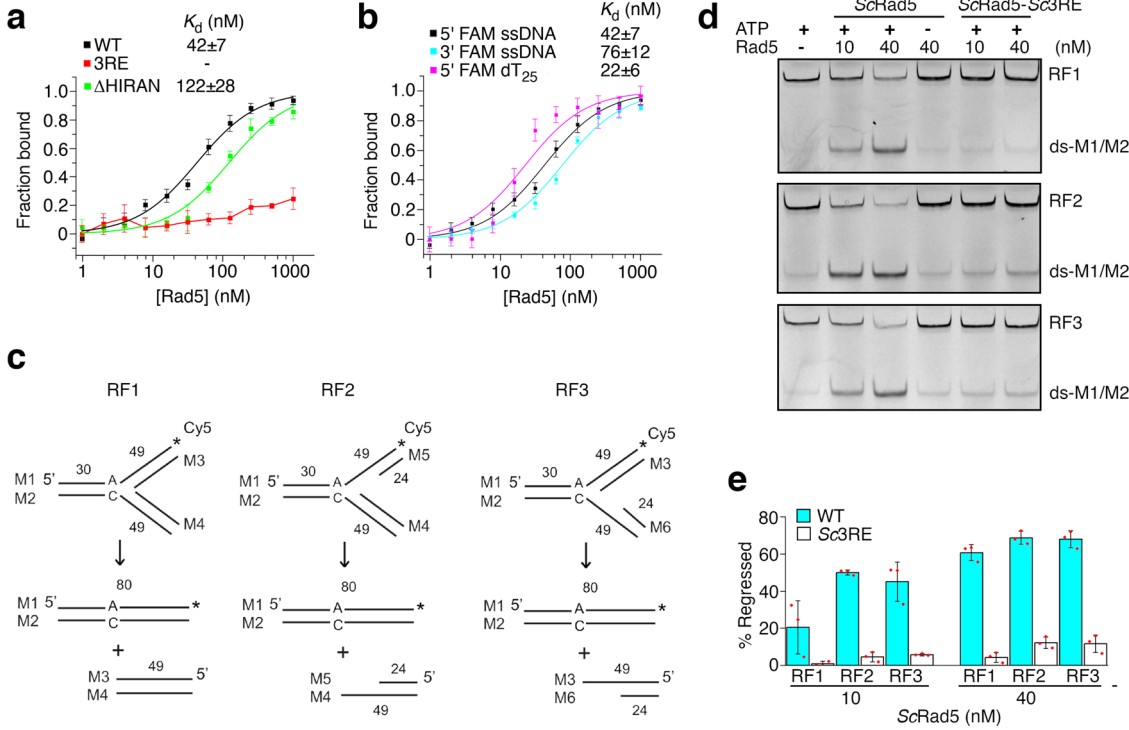

**Fig. 4 The conserved and positively charged region in Rad5's HIRAN domain mediates interactions with ssDNA and is required for the Rad5-catalyzed replication fork regression. a** FP experiments probing ssDNA binding to the wild-type *Kl*Rad5 and its 3RE and ΔHIRAN variants. **b** FP experiments probing interactions between *Kl*Rad5 and ssDNA molecules. Experiments with 5'- or 3'-fluorescein amidite (FAM)-labeled ssDNA molecules or 5'-FAM-labeled dT$_{25}$ are presented. **c** Reaction scheme for replication fork regression on the Cy5-labeled (*) movable replication fork-mimicking substrates. An A/C mismatch is engineered into the substrates to minimize spontaneous regression. RF1 contains no ssDNA gaps, RF2 and RF3 contain 25-nt ssDNA gaps in the leading and lagging arms, respectively. **d** Replication fork regression catalyzed by *Sc*Rad5 or its *Sc*3RE variant. **e** Quantification of the replication fork regression experiments. Data in panels **a**, **b**, and **e** are presented as mean values +/− standard deviations of three independent experiments. The red dots in panel **e** represent individual experiments. Source data for panels **a**, **b** and **d**, **e** are provided as Source Data file.

3'-hydroxyl group, we purified this domain in *Kl*Rad5 (*Kl*HIRAN) and *Sc*Rad5 (*Sc*HIRAN) and performed electrophoretic mobility shift assays (EMSA) with ssDNA with blocked 3'- or 5'-ends (Supplementary Fig. 8b). We found that *Kl*HIRAN and *Sc*HIRAN bind to both ssDNA molecules and do not possess any detectable preference toward ssDNA with an unblocked 3'-hydroxyl group. In contrast, a strong preference of *Hs*HIRAN towards ssDNA with an unblocked 3'-hydroxyl group is observed, consistent with previous reports[13–15]. Together, these data suggest that *Kl*Rad5's and *Sc*Rad5's HIRAN domains bind to ssDNA in a manner that does not require its 3'-hydroxyl group.

To further probe the role of *Kl*Rad5's HIRAN domain in DNA binding, we measured the affinity of *Kl*HIRAN to ssDNA and dsDNA with blunt ends or 5'- or 3'-overhanging ssDNA regions (Supplementary Fig. 8c). We found that *Kl*HIRAN binds to these types of DNA and its affinity to ssDNA and dsDNA with overhanging ssDNA regions are much stronger. Such DNA-binding property is similar to the DNA-binding property reported for HLTF's HIRAN domain[14]. Consistent with the role of *Kl*Rad5's HIRAN domain in binding dsDNA, we found that the ΔHIRAN truncation reduced its affinity to dsDNA (Supplementary Fig. 8d). The binding is also strongly inhibited by the 3RE substitution, suggesting that the positively charged region in *Kl*Rad5's HIRAN domain contributes to dsDNA binding. Consistent with these findings, we found that the 3RE substitution and the ΔHIRAN truncation inhibited the dsDNA-stimulated ATPase activity of *Kl*Rad5 (Supplementary Fig. 8e).

Our data suggest that the positively charged region in *Kl*Rad5's HIRAN domain contributes to binding both PCNA and DNA. To probe the interplay of DNA and PCNA binding to *Kl*Rad5, we

repeated the *Kl*Rad5-catalyzed PCNA poly-ubiquitination experiments in the presence of DNA. Inhibition of the reaction was observed for both dsDNA and ssDNA and the inhibition by ssDNA can be relieved by SSB (Supplementary Fig. 9a, b), which presumably binds to ssDNA and blocks the ssDNA–*Kl*Rad5 interaction. In contrast, neither ssDNA nor dsDNA inhibited the *Kl*Rad5-stimulated free ubiquitin-chain formation by Ubc13-Mms2 (Supplementary Fig. 9c, d). Together, these data suggest that DNA inhibits *Kl*Rad5-catalyzed PCNA poly-ubiquitination by competing with it for binding to *Kl*Rad5, but does not inhibit *Kl*Rad5's ubiquitin ligase activity per se. A stimulatory effect of DNA on the *Kl*Rad5-stimulated free ubiquitin-chain formation is observed (Supplementary Fig. 9d). Similar stimulatory effects by DNA have been observed for HLTF[30,31]. The mechanism of this stimulation requires further investigation. Interestingly, the severe defect in PCNA poly-ubiquitination caused by the 3RE substitution or the ΔHIRAN truncation was further aggravated by ssDNA or dsDNA (Supplementary Fig. 9e). These data suggest that regions outside of the HIRAN domain may also contribute to DNA's modulation on the *Kl*Rad5-catalyzed PCNA ubiquitination.

**The HIRAN domain contributes to the Rad5-catalyzed replication fork regression**. The ssDNA-binding site in HLTF's HIRAN domain plays a critical role in the replication fork regression it catalyzes[15,32]. To test if the equivalent region in Rad5 is also important for the Rad5-catalyzed replication fork regression, we first assessed the effect of the *Sc*3RE substitution on replication fork regression by *Sc*Rad5, which is well-established in the literature[5–7]. Fork-mimicking DNA substrates without

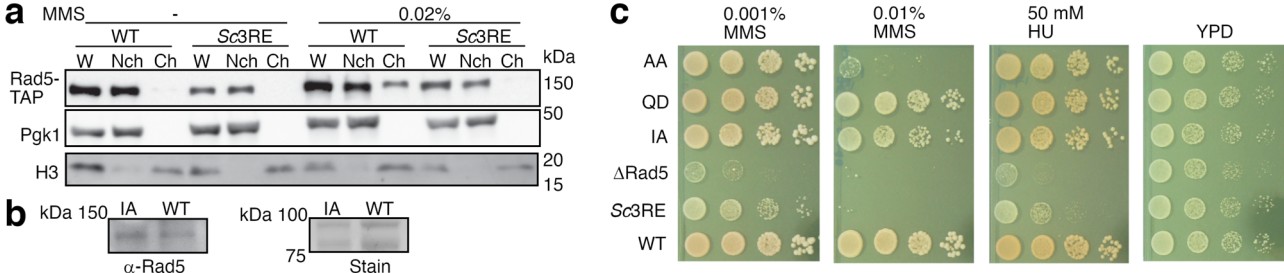

**Fig. 5 Physiological function of the conserved and positively charged region in Rad5's HIRAN domain. a** The *Sc*3RE substitution abolishes MMS-induced *Sc*Rad5–chromatin association. Fractions in chromatin sedimentation experiments are analyzed for *Sc*Rad5 with western blot. W whole-cell extract, Nch non-chromatin fraction, Ch chromatin fraction. Pgk1 and the histone protein H3 are included as markers for the non-chromatin and chromatin fractions, respectively. **b** The I916A substitution does not affect the *Sc*Rad5 protein level. The *Sc*Rad5 protein level is probed by western blot with an anti-Rad5 antibody. Equal loading was confirmed by Ponceau S staining (right panel). **c** The *Sc*3RE substitution causes sensitivity to replication stress-causing agents MMS and HU. AA, D681A/E682A; IA, I916A; QD, Q1061D; ΔRad5, Rad5 deletion; YPD, yeast extract peptone dextrose medium. For experiments presented in panels **a**–**c**, similar results were obtained using at least two independent strains per genotype. Source data for panels **a**, **b** are provided as Source Data file.

ssDNA gaps (RF1) or with 25-nt ssDNA gaps in the leading (RF2) or lagging (RF3) arms were tested (Fig. 4c). Strong fork regression activities toward all three substrates were observed, which were severely inhibited by the *Sc*3RE substitution (Fig. 4d, e). Using the same assay, we next probed the fork regression activity of *K. latics* Rad5. We found that it also possesses strong activity (Supplementary Fig. 10a, b), which severely inhibited by the Q1051D substitution, consistent with the previous report that fork regression by Rad5 requires ATP hydrolysis by its Snf2 domain[5]. Similar to *Sc*Rad5, fork regression by *K. lactis* Rad5 was also severely inhibited by the 3RE substitution (Supplementary Fig. 10a, b). Together, these data suggest that the putative DNA-binding site in Rad5's HIRAN also plays a critical role in the Rad5-catalyzed replication fork regression.

**Physiological functions of Rad5's HIRAN domain.** To assess the physiological function of Rad5's HIRAN domain in cells, we examined the effects of the *Sc*3RE substitution in budding yeast. Previous studies have indicated that *Sc*Rad5 forms sub-nuclear foci in replication-stressed conditions[9,10,33,34], which likely reflects the recruitment of *Sc*Rad5 to DNA damage sites or stressed replication forks[9,10]. Consistent with these reports, our chromatin fractionation experiments revealed a strong *Sc*Rad5–chromatin association in MMS-treated cells (Fig. 5a). Remarkably, we found that the *Sc*3RE substitution abolished the *Sc*Rad5–chromatin association (Fig. 5a). This is in line with a previous report that the *Sc*Rad5 foci formation is dependent on its HIRAN domain[9], and further suggests that the positively charged region in Rad5's HIRAN domain plays a critical role in targeting Rad5 to stressed replication forks or DNA lesions.

We next examined the effects of the *Sc*3RE substitution on cells' sensitivity toward replication stress-causing agents MMS and hydroxyurea (HU), as Rad5 null mutations are sensitive to these drugs[10,35]. For comparison, we also included the I916A[27] and the Q1061D[36] substitutions that impair *Sc*Rad5's ubiquitin ligase and ATPase activities, respectively; and the D681A/E682A substitution that impairs both activities[36]. We have previously shown that the D681A/E682A and Q1061A substitutions do not affect the *Sc*Rad5 protein level[36]. We found that this is also the case for the I916A substitution (Fig. 5b). Our data indicated that the *Sc*3RE substitution causes sensitivity towards MMS and HU at a level much greater than those caused by the I916A, the Q1061D, or the D681A/E682A substitutions (Fig. 5c). It has been reported that disrupting the *Sc*Rad5–Rev1 interaction by the F13A/N14A substitution caused comparable or weaker sensitivity towards MMS or HU compared to the sensitivity caused by the

I916A substitution[8,10]. The much stronger sensitivity we observed for the *Sc*3RE substitution suggests that it impairs multiple activities of *Sc*Rad5 in vivo.

**The Snf2–HIRAN interaction contributes to Rad5's multiple activities.** In our structure, the HIRAN domain forms extensive interactions with the Snf2 domain. The interactions are largely mediated by a "mediator helix" located between these domains (Fig. 1c), which overlaps with the leucine heptad repeat identified in *Sc*Rad5[35]. The β1–β3 insertion present in the Snf2 domain lobe 1 and α1, β1 and β5–6 in the HIRAN domain also play critical roles in the Snf2–HIRAN interaction. The HIRAN-mediator helix, Snf2-mediator helix, Snf2–HIRAN interfaces bury 1000 Å², 1100 Å². and 1000 Å² of surface area. An extensive hydrophobic core is formed by residues in the C-terminal half of the mediator helix, the β1–β3 insertion in the Snf2 domain lobe 1, and the HIRAN domain (Supplementary Fig. 11a). The mediator helix also forms additional and largely hydrophobic interactions with the HIRAN (Supplementary Fig. 11b) and Snf2 (Supplementary Fig. 11c) domains. Previous sequence analysis has suggested that the β1–β3 insertion in the Snf2 domain lobe 1 is unique for the Rad5 family of Snf2 enzymes[37]. Our structure suggests that a function of this unique element is connecting the HIRAN and Snf2 domains.

The extensive Snf2–HIRAN interaction suggests that it has an important role in stabilizing the overall Rad5 structure and contributes to Rad5's multiple activities. To probe its function, we examined how disrupting this interaction affects *Kl*Rad5's multiple activities. We generated *Kl*Rad5 variants with alanine substitutions at Ile322, Met323, Leu325, or Phe326 in the mediator helix that mediate extensive interactions with the Snf2 domain and/or the HIRAN domain (Supplementary Fig. 11a, b) and a full-length *K. lactis* Rad5 variant without the β1–β3 insertion in the Snf2 domain lobe 1 (Δβ1–3). A 6x histidine tag (Histag) was introduced to the N-terminus of these variants and the wild-type *Kl*Rad5 to facilitate their detection by western blotting. Size-exclusion chromatography indicated that the Histag did not affect the overall folding of *Kl*Rad5, and the predominant species of these *K. lactis* Rad5 variants were correctly folded (Supplementary Fig. 2b). Minor species with larger molecular sizes were observed for the L325A, F326A, and Δβ1–3 variants, consistent with decreased protein stability. Activity experiments indicated that the Histag-*Kl*Rad5 possesses robust dsDNA-stimulated ATPase activity (Supplementary Fig. 11d), dsDNA-, ssDNA-, and PCNA-binding activity (Supplementary Fig. 11e–g) and free and PCNA-anchored ubiquitin-chain extension activity

(Supplementary Fig. 11h, i), which were strongly suppressed by the substitutions or the Δβ1–3 truncation. Together, these data suggest that the Snf2–HIRAN interaction contributes to multiple activities of KlRad5. The reduced activity in catalyzing the PCNA-anchored ubiquitin-chain extension possessed by the I322A, M323A, and L325A variants was further suppressed by ssDNA or dsDNA (Supplementary Fig. 11i), consistent with the notion that the HIRAN domain and additional regions in KlRad5 contribute to DNA's modulation on the KlRad5-catalyzed PCNA ubiquitination.

## Discussion

Our study provides insights into the coordination among Rad5's domains to perform its multiple activities. First, our data suggest that the HIRAN domain recruits PCNA and facilitates the PCNA-anchored ubiquitin-chain extension by the RING domain. Our structure shows that the RING domain has minimal contact with the rest of KlRad5 (Fig. 1b) and a fully exposed active site (Supplementary Figs. 3d–f), suggesting that it is mobile and has an autonomous function. Such properties of the RING domain are consistent with our previous genetic finding that the ubiquitin ligase and ATPase activities of ScRad5 contribute separately to replication stress tolerance[36], and could provide the flexibility to accommodate increasing numbers of ubiquitin moieties during PCNA-anchored ubiquitin-chain extension. Although en bloc ubiquitin-chain transfer to PCNA has been reported[31], recent studies suggested that in physiological environments Rad5 or HLTF catalyzes PCNA poly-ubiquitination by extending the PCNA-anchored ubiquitin chain[28,30]. Consistently, our data indicated that the Sc3RE substitution inhibited PCNA poly-ubiquitination in MMS-stressed cells (Fig. 3f). Together, our data suggest that the HIRAN–RING coordination takes place in vitro and in vivo, and plays a critical role in initiating the error-free DDT. Second, our data suggest a critical role of the HIRAN domain in replication fork regression, in which the Snf2 domain serves as the motor[5]. Replication fork regression has been observed in eukaryotes including yeast and human, particularly in replication-stressed conditions[6,7,38]. Rad5 and Mph1 are the major enzymes in budding yeast with confirmed replication fork regression activity[7]. We have previously shown that ScRad5's ATPase activity that drives replication fork regression makes a separated contribution to replication stress tolerance[36]. These previous studies suggest an important function of the Rad5-catalyzed replication fork regression in replication stress tolerance in yeast. The HIRAN–Snf2 coordination in this reaction merits further investigation. Our data indicated that neither KlRad5's nor ScRad5's HRINA domain forms specific interactions with the ssDNA 3'-hydroxyl group, which is critical for the HLTF-catalyzed fork regression[15,32]. These data suggest that although HLTF's and Rad5's HIRAN domains are both important for replication fork regression, they probably have distinct functions in this reaction. Recent studies suggest that the propensities and physiological functions of replication fork regression in yeast and human cells are vastly different[6,7,38]. It is interesting to investigate how the distinct replication fork regression mechanisms by HLTF and Rad5 with respect to the HIRAN-ssDNA interaction contribute to these differences.

Our biochemical data suggest that the conserved and positively charged region in Rad5's HIRAN domain plays critical roles in binding PCNA and facilitating its ubiquitination, binding DNA and in the Rad5-catalyzed replication fork regression. Our cellular experiments also suggest that this region is critical for targeting Rad5 to stressed replication forks or DNA lesions and contributes to Rad5's multiple activities in vivo. Collectively, these data suggest that this region is the linchpin of Rad5 that plays critical

roles in its multiple activities. Highlighting the functional importance of this region, it was recently reported that the R187E substitution in this region in ScRad5 (equivalent to R190E in KlRad5) eliminated fitness defects caused by ScRad5 overexpression[39].

By serving as an interaction hub, the positively charged region in Rad5's HIRAN domain may play a role in coordinating its multiple activities. Our in vitro experiments suggest that DNA competes with PCNA for binding to this region and inhibits the KlRad5-catalyzed PCNA poly-ubiquitination. A similar dsDNA-mediated inhibition has been observed for the ScRad5-catalyzed PCNA poly-ubiquitination[28]. In this study and ours, the ubiquitin–PCNA fusion protein was used as the substrate for Rad5, which was not loaded to DNA. The physiological relevance of the observed inhibition is unclear since PCNA is loaded on dsDNA during replication. Future studies are required to address how DNA modulates the Rad5-catalyzed PCNA poly-ubiquitination in the physiological environment. Likewise, the loaded PCNA may modulate the HIRAN–DNA interaction to affect Rad5's recruitment to stressed replication forks or DNA lesions and/or the Rad5-catalyzed fork regression. Further studies are required to investigate this possibility.

Our study reveals both similarities and differences between the function of Rad5's HIRAN domain and the previously characterized HIRAN domain in HLTF[13–15,32]. First, our data suggest that Rad5's but not HLTF's HIRAN domain mediates interactions with PCNA and promotes its poly-ubiquitination. Second, our data indicate that although both Rad5's and HLTF's HIRAN domain bind ssDNA, they bind ssDNA with different mechanisms. Third, we found that although HLTF's and Rad5's HIRAN domains are essential for the replication fork regression they catalyze, they probably have distinct functions in this reaction. Sequence analysis has suggested that HIRAN domains in HLTF and Rad5 represent two major sub-family of HIRAN domains in eukaryotes[40]. Our study suggests that these sub-families possess distinct functions. The structural difference between HLTF's and Rad5's HIRAN domains, especially at the positively charged region, probably contributes to the differences in their function. In addition to this region, our data indicate that the HIRAN–Snf2 interaction is also critical for Rad5's multiple activities. The HIRAN domain plays critical roles in several of these activities, including PCNA binding and ubiquitination, and DNA binding. Our structure suggests large differences between the HIRAN–Snf2 interaction in Rad5 and HLTF since the mediator helix in our structure and its equivalent in isolated HLTF N-terminal region structures[13–15] adopt drastically different conformations (Supplementary Fig. 4a), and the β1–β3 insertion in the Snf2 domain lobe 1 is poorly conserved between Rad5 and HLTF (Supplementary Fig. 1). Together, these observations suggest that the difference in the HIRAN–Snf2 interaction probably also contributes to the differences in the function of the HIRAN domain in Rad5 and HLTF.

In summary, our work illuminated the spatial arrangement of the different domains in Rad5 and defined a new type of HIRAN domain that is widely distributed in eukaryotes, which contributes to Rad5's multiple activities. Our study provides insights into the functional coordination among Rad5's domains and the molecular mechanism of Rad5's function in replication-stressed contexts.

## Methods

**Protein expression and purification**. The KlRad5 expression plasmid was constructed by inserting the gene fragment corresponding to the K. lactis Rad5 residues 163–1114 into the vector pET26b (Novagen). E. coli BL21 Rosetta (DE3) cells transformed with this plasmid were cultured in LB medium supplemented with 34 mg/l kanamycin and 25 mg/l chloramphenicol and induced with 0.3 mM

Isopropyl β-D-1-thiogalactopyranoside (Bio Basic) at 16 °C for 14 h. Collected cells were resuspended in a buffer containing 20 mM Tris (pH 7.5), 300 mM sodium chloride, and 2 mM β-mercaptoethanol (β-ME) and lysed by an AH-2010 homogenizer (ATS Engineering). Although the recombinant KlRad5 does not contain an affinity tag, it binds to nickel–nitrilotriacetic acid (Ni-NTA) agarose resin (Qiagen). After washing the resin four times with 20 bed volumes of the same buffer supplemented with 5 imidazole, KlRad5 was eluted with the same buffer supplemented with 200 imidazole. KlRad5 was further purified by Heparin (Hitrap Heparin HP, GE Healthcare) and ion-exchange (Hitrap Q HP, GE Healthcare) columns with a 0 to 1 M sodium chloride gradient in a buffer containing 20 mM Tris (pH 7.5) and 2 mM dithiothreitol (DTT), and a size-exclusion (Sephacryl S300 HR or Superose 6 10/300, GE Healthcare) column with a buffer containing 20 mM Tris (pH 7.5), 200 mM sodium chloride, and 2 mM DTT, concentrated to 10 mg/ml, flash-cooled in liquid nitrogen and stored at −80 °C.

To construct the expression plasmid for the full-length K. lactis Rad5 (KlRad5-FL), codon usage for the first 162 residues in K. lactis Rad5 was optimized for expression in E. coli. The optimized gene fragment was chemically synthesized (Supplementary Table 1) (Sangon Biotech) and inserted into the KlRad5 expression plasmid. To construct the expression plasmid for KlRad5 with the Δβ1–3 deletion, the gene fragment coding for residues 412–478 in the KlRad5-FL expression plasmid was replaced with an oligonucleotide coding for a G–S–G–S peptide. To construct expression plasmids for KlRad5 variants with an N-terminal Histag, the related gene fragments were inserted into the vector pET28a. The interactions among the HIRAN domain, the mediator helix, and the β1–β3 insertion in the Snf2 domain lobe 1 are primarily hydrophobic in nature (Supplementary Fig. 11a, b), indicating that removing the HIRAN domain or the HIRAN domain and the mediator helix will disrupt the protein folding by exposing a large number of hydrophobic residues. Indeed, we found that removing the HIRAN domain and the mediator helix from KlRad5 made it insoluble. To construct a correctly folded variant of K. lactis Rad5 without the HIRAN domain (ΔHIRAN), we removed residues 1–353 encompassing the HIRAN domain and mediator helix and replaced the β1–β3 insertion in the Snf2 domain lobe 1 (residues 412–478) with a G–S–G–S linker. The modified gene fragment was inserted into the vector pET28a. To construct plasmids for hemagglutinin (HA) tagged K. lactis Rad5 variants, an oligonucleotide coding the HA peptide was inserted into plasmids for the K. lactis Rad5 variants. The above K. lactis Rad5 variants were expressed and purified following the same protocol for KlRad5. To construct the expression plasmid for the flag-tagged full-length K. lactis Rad5 (flag-KlRad5-FL), an oligonucleotide coding the flag peptide was inserted into the KlRad5-FL expression plasmid. Flag-KlRad5-FL expressed in E. coli BL21-CodonPlus (DE3)-RIP cells (Agilent) was bound to Ni-NTA agarose resin (ThermoFisher) in a buffer containing 20 mM potassium phosphate (pH 7.4), 10% glycerol, 0.5 mM EDTA, 0.01% NP-40, 1 mM DTT, and 150 mM potassium chloride. The resin was washed with 10 bed volumes of the same buffer supplemented with 0.1% NP-40 and 5 mM imidazole, and flag-KlRad5-FL was eluted with the same buffer supplemented with 200 mM imidazole. Ni-NTA-purified flag-KlRad5-FL was subsequently bound to anti-FLAG M2 agarose resin (Sigma-Aldrich). After washing the resin with 10 bed volumes of the same buffer supplemented with 0.1% NP-40, flag-KlRad5-FL was eluted with the same buffer supplemented with 200 μg/ml of FLAG peptide and dialyzed against the same buffer. To construct expression plasmids for KlHIRAN and ScHIRAN, gene fragments corresponding to the K. lactis Rad5 residues 174–341 or the S. cerevisiae Rad5 residues 171–375 were inserted into the vector pTXB1 (New England Biolabs). KlHIRAN and ScHIRAN fused to the intein-chitin-binding-domain tag expressed in E. coli BL21 Rosetta (DE3) cells were bound to chitin resin (New England Biolabs) in a buffer containing 20 mM Tris (pH 7.5) and 300 mM sodium chloride. After washing the resin three times with 20 bed volumes of the same buffer, the resin was incubated with the same buffer supplemented with 200 mM DTT for 12 h at 4 °C to cleave the intein-chitin-binding-domain tag. KlHIRAN and ScHIRAN were eluted from the resin with the same buffer and further purified by a size-exclusion column (Superdex S200 Increase 10/300, GE Healthcare). The expression plasmid for his-flag-PCNA was constructed by inserting an oligonucleotide coding the flag peptide and the K. lactis PCNA gene into the vector pET28a. The expression plasmid for the his-flag-ubiquitin–PCNA fusion protein was constructed by inserting the S. cerevisiae ubiquitin gene and an oligonucleotide coding for a VQIPGK linker into the his-flag-PCNA expression plasmid, between sequences for the flag peptide and PCNA. The expression plasmid for the his-flag-tagged human PCNA (his-flag-HsPCNA) was constructed by inserting an oligonucleotide coding the flag peptide and the Homo sapiens PCNA gene into the vector pET28a. His-flag-PCNA, the his-flag-ubiquitin–PCNA fusion protein and his-flag-HsPCNA were purified by Ni-NTA, ion-exchange (Hitrap Q HP), and size-exclusion (Sephacryl S100 HR, GE Healthcare) columns. The expression plasmids for his-tagged and untagged ubiquitin were constructed by inserting the S. cerevisiae ubiquitin gene into plasmids pET28a and pTYB2 (New England Biolabs), respectively. The his-tagged ubiquitin was purified by Ni-NTA, ion-exchange (Hitrap Q HP), and size-exclusion (Sephacryl S100 HR) columns. The untagged ubiquitin was purified by chitin (New England Biolabs), ion-exchange (Hitrap Q HP), and size-exclusion (Sephacryl S100 HR) columns. The expression plasmids for Ubc13 and Mms2 were constructed by inserting the K. lactis Ubc13 and Mms2 genes into vectors pET26b and pTYB2, respectively. The plasmids were co-transformed into E. coli BL21 Rosetta (DE3) cells and the Mms2-Ubc13 complex was purified by chitin, ion-exchange (Hitrap Q HP), and

size-exclusion (Sephacryl S100 HR) columns. The S. cerevisiae ubiquitin-activating enzyme Uba1 (ScUba1) was expressed and purified as described[41]. Briefly, the gene fragment corresponding to the S. cerevisiae Uba1 residues 10–1024 was inserted into vector pET28a. The plasmid was transformed into E. coli BL21 Rosetta (DE3) cells for protein expression. ScUba1 was purified by Ni-NTA, hydrophobic interaction column (Hitrap Butyl HP, GE Healthcare) with a 900 to 0 mM ammonium sulfate gradient in a buffer containing 20 mM Tris (pH 7.5) and 2 mM DTT and a size-exclusion column (Superdex S200 Increase 10/300). ScRad5 was expressed with a modified pYES2-His-ScRad5 plasmid[28], in which the N-terminal 6x histidine tag was replaced with a flag peptide, in the protease-deficient yeast strain 334 (MATα pep4-3prb1-1122 ura3-52 leu2-3,112 reg1-501 gal1, Supplementary Table 2). Cells were cultured in the uracil omission medium with 2% glucose at 30 °C until OD_{660} reaches 0.8 and induced by 2% galactose for 12 h. To purify ScRad5, cells were lysed in the K buffer (20 mM potassium phosphate (pH 7.4), 10% glycerol, 0.5 mM EDTA, 0.1% NP-40, 1 mM β-ME) supplemented with 5 μg/ml of aprotinin, chymostatin, leupeptin, and pepstatin A, 1 mM phenylmethylsulfonylfluoride, and 300 mM potassium chloride, and the cleared lysate was loaded to anti-flag M2 resin (Sigma-Aldrich). After washing the resin with 80 bed volumes of K buffer supplemented with 150 mM potassium chloride and 0.1% NP-40 and 50 bed volumes of K buffer supplemented with 150 mM potassium chloride and 0.01% NP-40, ScRad5 was eluted with K buffer supplemented with 150 mM potassium chloride and 200 μg/ml FLAG peptide. To construct the expression plasmid for HLTF, the gene for the Homo sapiens HLTF was optimized for expression in E. coli and synthesized (Supplementary Table 1, Sangon Biotech) and the fragment corresponding to residues 25–1013 was inserted into the vector pET28a. To construct the expression plasmid for HA-tagged HLTF, an oligonucleotide coding the HA peptide was inserted into the HLTF plasmid. HLTF was purified by Ni-NTA, ion-exchange (Hitrap SP HP), and size-exclusion (Superdex S200 Increase 10/300) columns. To construct the expression plasmid for the HA-tagged HLTF HIRAN domain (HsHIRAN), an oligonucleotide coding the HA peptide and the gene fragment corresponding to HLTF residues 55–180 were inserted into the vector pET28a. The HA-tagged HsHIRAN was purified by Ni-NTA and size-exclusion (Superdex S200 Increase 10/300) columns. Unless otherwise indicated, the buffers used in the Ni-NTA, ion exchange, and size-exclusion purification steps were the same as the corresponding buffers for the KlRad5 purification; the buffers used in the chitin purification steps were the same as the buffers used in the same step for the KlHIRAN and ScHIRAN purification. All proteins were flashed cooled in liquid nitrogen and stored at −80 °C prior to use.

Amino acid substitutions were generated with polymerase chain reactions following instructions of the QuikChange kit (Agilent Technologies) and verified by DNA sequencing. The expression and purification of the substituted proteins followed the same protocol for the wild-type proteins.

Primers used for expression plasmid construction and amino acid substitution are listed in Supplementary Table 3.

**Crystallization and crystal structure determination.** KlRad5 (10 mg/ml) crystallized with a sitting-drop setup at 20 °C. The reservoir liquid contains 1.0 M ammonium citrate tribasic (pH 7.0) and 0.1 M Hepes (pH 7.6). Before data collection, crystals were equilibrated in the reservoir solution supplemented with 30% glycerol for 5 s, flash-cooled, and stored in liquid nitrogen. To generate mercury-derivatized crystals, crystals were equilibrated in the reservoir solution supplemented with 30% glycerol and 6 mM mercury potassium iodide for 15 s, flash-cooled, and stored in liquid nitrogen. The crystals belong to space group P6₂22 and contain one KlRad5 molecule in the asymmetric unit. Diffraction data were collected on an ADSC Q315 charge-coupled device detector at the Shanghai Synchrotron Radiation Facility beamline BL17U, at 100 K. A single-wavelength anomalous diffraction (SAD) dataset was generated by merging diffraction data collected on three mercury-derivatized crystals near the mercury K-edge (1.009 Å). Another dataset was collected on a native crystal at 0.9796 Å. Diffraction data were scaled and merged with HKL2000[42]. Phases for the reflections were determined with PHENIX[43] with the SAD dataset. The structure was built and modified with COOT[44] and O[45]. Structure refinement was carried out with PHENIX. Structures of the mercury-derivatized and native crystals were refined to resolutions of 3.3 Å and 3.6 Å, respectively (Table 1). Minor differences between these structures were observed. Residues 163–173, 208–223, 296–308, 421–430, 445–453, 515–538, and 796–826 were disordered in the mercury-derivatized crystal. Additional residues 294–295, 309–310, and 336–359 were not included in the structure for the native crystal due to weak electron density. In the structure for the mercury-derivatized crystal, three mercury ions are bound to the protein, two of which replace the zinc ions in the RING domain. Structural homologs were identified with the Dali server[46].

**Negative staining electron microscope.** In total, 10 μl of purified KlRad5 (5 μg/ml) was applied to a glow-discharged EM carbon grid and stained with Nano-W (Nanoprobes). The EM grids were imaged on a FEI Tecnai T12 microscope operated at 120 kV. Images were recorded at a nominal magnification of 67,000×, with a 4 k × 4 k Eagle CCD camera, corresponding to 1.74 Å per pixel on the specimen. EMAN2.1[47] was used for EM reconstruction. 10,116 particles were selected with e2boxer.py, the 2D averaging classes were calculated with e2refine2d.

py. The initial model was generated using e2initialmodel.py. Model refinement was carried out with e2refine_easy.py. The estimated resolution of the reconstruction based on the gold-standard criterion[48] is 25 Å.

**Dynamic light scattering**. Dynamic light scattering experiments were performed on a DynaPro NanoStar instrument (Wyatt Technologies) at 25 °C. $Kl$Rad5 was characterized at a concentration of 1 mg/ml, in a buffer containing 20 mM Tris (pH 7.5), 200 mM sodium chloride and 2 mM DTT. Data were analyzed with the DYNAMICS V6 software (Wyatt Technologies).

**Hydrogen-deuterium exchange mass spectrometry**. The dsDNA molecule used in hydrogen-deuterium exchange was generated by heating a self-annealing oligonucleotide D1 (Supplementary Table 4, synthesized by Sangon Biotech) to 95 °C followed by slow cooling to 4 °C. For deuterium labeling, $Kl$Rad5 (7.5 mg/ml) in a buffer containing 20 mM Tris (pH 7.5), 200 mM sodium chloride in the absence or presence of 90 μM dsDNA was diluted tenfold with the labeling buffer (20 mM Tris (pD 7.1), 200 mM sodium chloride, 100% $D_2O$). After incubation at 25 °C for 60 or 300 s, deuterium uptake was quenched by adding equal volume of ice-cold quenching buffer (4 M guanidine hydrochloride, 200 mM citric acid, and 500 mM tris(2-carboxyethyl)phosphine, 100% $H_2O$, pH 1.8). After 5-min digestion with 0.1 μM pepsin (Promega) and Protease from *Aspergillus saitoi* Type XIII (Sigma-Aldrich) on ice, the sample was cleared by centrifugation, and resulting peptides were separated by an ACQUITY UPLC BEH C4 column (particle size 1.7 μm, column dimensions 2.1 × 50 mm, Waters) with an Ultimate 3000 UPLC system (Thermo Scientific). Mass spectrometry analysis of the peptides was performed on a Q Exactive mass spectrometer (Thermo Scientific). Mass spectrometry data were matched to peptides in $Kl$Rad5 with PROTEOME DISCOVERER (Thermo Scientific). Peptide peaks were inspected with XCALIBUR (Thermo Scientific). To estimate the maximum deuterium uptake of the peptides, the above experiments were repeated with an extended incubation in $D_2O$ for 24 h. Deuterium uptake levels were calculated with HDExaminer (Sierra Analytics). Three repeats of each experiment were performed.

**ATPase assay**. The dsDNA molecule used in the ATPase assay was generated by heating complimenting oligonucleotides D2a and D2b (Supplementary Table 4, synthesized by Sangon Biotech) to 95 °C followed by slow cooling to 4 °C. ATPase activity was measured by coupling ADP production to NADH oxidation with pyruvate kinase and lactate dehydrogenase[49]. The resulting absorption change at 340 nm was monitored on an ultraspec 2100 pro spectrophotometer (GE Healthcare). The reaction mixture contains 0.25 μM $Kl$Rad5, 40 mM Tris (pH 7.5), 50 mM sodium chloride, 0.5 mM ATP, 1 mM phosphoenolpyruvate, 0.2 mg/ml NADH, 80 units/ml pyruvate kinase, 100 units/ml lactate dehydrogenase, 10 mM magnesium chloride, and dsDNA at indicated concentrations. Three repeats of each experiment were performed. Data were analyzed with QTIPLOT (www.qtiplot.com).

**DNA co-precipitation**. Biotin-labeled oligonucleotides were synthesized by Sangon Biotech (for HLTF and $Kl$Rad5 co-precipitation) or Integrated DNA Technologies (for $Sc$Rad5 co-precipitation). A biotin-labeled dsDNA molecule was generated by heating a 5′-biotin-labeled self-annealing oligonucleotide D1 (Supplementary Table 4) to 95 °C followed by slow cooling to 4 °C. To characterize DNA binding to $Kl$Rad5 or HLTF, a 100 μl binding mixture containing the binding buffer, 2 μM $Kl$Rad5 or HLTF and 1 μM of the biotin-labeled dsDNA or $dT_{25}$ with 5′- or 3′-biotin labels was incubated with 10 μl SoftLink Soft Release avidin resin (Promega) for 1 h at 4 °C. The binding buffer contains 20 mM Tris (pH 7.5) and 50 mM sodium chloride unless indicated otherwise. After washing the resin three times with the binding buffer, bound DNA and protein were eluted with the binding buffer supplemented with 5 mM biotin and analyzed with sodium dodecyl sulfate (SDS) polyacrylamide gel electrophoresis (PAGE). When indicated, SSB (Sangon Biotech) was added to a final concentration of 20 μM. To characterize DNA binding to $Sc$Rad5, 0.15 μM $Sc$Rad5 and 0.2 μM $dT_{25}$ with 5′- or 3′-biotin labels were incubated with 10 μl Streptavidin Magnetic Beads (NEB) in a 100 μl binding mixture with the same binding buffer. SSB was added to a final concentration of 1 μM when indicated.

**Fluorescence polarization**. Fluorescein amidite (FAM) labeled and unlabeled oligonucleotides were synthesized by Sangon Biotech. Their sequences are listed in Supplementary Table 4. The dsDNA molecule for the FP experiments with $Kl$Rad5 was produced by heating a mixture of 5′-FAM-labeled oligonucleotides D2a/b to 95 °C, followed by slow cooling to 4 °C. The ssDNA molecule for the FP experiments with $Kl$Rad5 contains the S1 sequence and a 5′-FAM label unless indicated otherwise. To characterize DNA binding to $Kl$Rad5, 1 nM of FAM-labeled ds- or ssDNA was mixed with $Kl$Rad5 at indicated concentrations in a buffer containing 20 mM Tris (pH 7.5), 50 mM (for dsDNA binding), or 70 mM (for ssDNA binding) sodium chloride and 2 mM magnesium chloride. After a 15-min incubation at room temperature, FP data were collected on a Synergy HT microplate reader (Biotek).

The dsDNA molecules for the FP experiments with $Kl$HIRAN were produced by annealing a 5′-FAM-labeled oligonucleotide with the D3a sequence to oligonucleotides with the D3b (blunt end dsDNA) or D3c (dsDNA with a 3′-overhanging ssDNA region) sequences, or a 3′-FAM-labeled oligonucleotide with the D3a sequence to a nucleotide with the D3d sequence (dsDNA with a 3′-overhanging ssDNA region). The ssDNA molecule for the FP experiments with $Kl$HIRAN contains the D3a sequence and a 5′-FAM label. The binding mixture contains 1 nM of FAM-labeled ds- or ssDNA, 20 mM Tris (pH 7.5), 50 mM sodium chloride, 2 mM magnesium chloride, and $Kl$HIRAN at indicated concentrations. After a 15-minute incubation at room temperature, fluorescence polarization data were collected on a Spark multimode microplate reader (TECAN).

The excitation and emission wavelengths for the FP experiments were 485 nm and 528 nm, respectively. All experiments were repeated three times. Data were analyzed with QTIPLOT.

**Electrophoretic mobility shift assay**. EMSA experiments were performed with isolated HIRAN domains and $dT_{10}$ oligonucleotides end-labeled with FAM (Sangon Biotech). The reaction mixtures contain 100 nM FAM-labeled $dT_{10}$, 20 mM Tris (pH 7.5), 100 mM potassium chloride, 0.5 mM DTT, 3% glycerol, 2 mM magnesium chloride, 250 μg/mL BSA, and isolated HIRAN domains at indicated concentrations. After 20-min incubation at 4 °C, the binding reactions were analyzed with a native 10% acrylamide gel. Gels were imaged with the ChemiDoc Touch imaging system (Bio-rad).

**PCNA co-precipitation**. To characterize the PCNA-Rad5 interaction, a 90 μl of a binding mixture containing the binding buffer, 2 nmol his-flag-PCNA or its variants, and 2 nmol $Kl$Rad5 or its variants was incubated with 40 μl anti-flag M2 affinity resin (Sigma-Aldrich) at 4 °C for 2 h. The binding buffer contains 20 mM Tris (pH 7.5), 100 mM sodium chloride, and 1 mM DTT. After washing the resin four times with the binding buffer, bound proteins were eluted with the binding buffer supplemented with 0.2 mg/ml 3×flag peptide (Sigma-Aldrich). To characterize the interaction between PCNA and HLTF or its HIRAN domain, his-flag-PCNA and $Kl$Rad5 were replaced by his-flag-$Hs$PCNA and HLTF or $Hs$HIRAN, respectively. The $Kl$Rad5 variants, HLTF, and $Hs$HIRAN used in experiments presented in Fig. 3a and Supplementary Fig. 7d, f contain an N-terminal HA tag. The eluted proteins were analyzed with western blot with anti-flag (a8592, Sigma-Aldrich, 1:5000 diluted) and anti-HA (ab18181, Abcam, 1:5000 diluted) antibodies. The $Kl$Rad5 variants used in the experiment presented in Supplementary Fig. 11g contain an N-terminal Histag. The eluted proteins were analyzed with western blot with an anti-Histag (ab1269, Abcam, 1:5000 diluted) antibody. Three repeats of the experiment were performed.

**Unanchored and PCNA-anchored ubiquitin-chain extension**. Unanchored ubiquitin-chain was produced in a reaction mixture containing 40 mM Tris (pH 7.5), 50 mM sodium chloride, 10 mM magnesium chloride, 1 mM ATP, 0.15 μM $Sc$Uba1, 2 μM Ubc13-Mms2 complex, 10 μM ubiquitin and 0.5 μM $Kl$Rad5, or its variants unless otherwise indicated. The reactions were allowed to proceed for 10 m at 30 °C and analyzed by Sodium dodecyl sulfate-polyacrylamide gel electrophoresis (SDS PAGE) with Coomassie blue staining. To quantify the activity, 10 μM His-tagged ubiquitin with the G76C substitution and 10 μM ubiquitin with the K63R substitution were used instead of the wild-type ubiquitin. Only di-ubiquitin can be produced in this reaction. The intensity of the corresponding band in the SDS PAGE was measured with ImageJ (imagej.nih.gov/ij/) and used to represent the activity. PCNA-anchored ubiquitin-chain was produced by supplementing the reaction with 0.05 μM his-flag-ubiquitin–PCNA fusion protein. The reactions were analyzed with western blot with an anti-flag antibody (a8592, Sigma-Aldrich, 1:5000 diluted). To quantify the activity, 10 μM ubiquitin with the K63R substitution was used instead of the wild-type ubiquitin. Only one ubiquitin molecule can be added to the his-flag-ubiquitin–PCNA fusion protein in this reaction. The intensity of bands corresponding to ubiquitinated and unmodified his-flag-ubiquitin–PCNA molecules in the western blot was read by ImageJ, and the fraction of his-flag-ubiquitin–PCNA molecules ubiquitinated was calculated to represent the activity. To assess the effect of DNA in ubiquitin-chain extension, ssDNA with the S1 sequence or dsDNA produced by annealing oligonucleotides with the D2a and D2b sequences (Supplementary Table 4) were added to the reaction to a final concentration of 10 μM. SSB was added to a final concentration of 80 μM when indicated. To better assess the weak activity of the 3RE, ΔHIRAN variants of $Kl$Rad5 and variants with disrupted HIRAN–Snf2 interactions, the concentration of $Sc$Uba1, the Ubc13-Mms2 complex, and $Kl$Rad5 were doubled in the reaction mixture (Supplementary Fig. 9e and 11i). Three repeats of each experiment were performed.

**Fork regression assay**. Substrates RF1-3 mimicking DNA replication forks were prepared by hybridizing oligonucleotides M1-6 (Supplementary Table 4, synthesized by Integrated DNA Technology) as described[50]. M1 was 3′ labeled with TdT (NEB) and Cy5-dUTP (Enzo Life Sciences). To construct RF1, the M1/M3 and M2/M4 pairs were annealed first in NEB Buffer 3.1, the annealed pairs were subsequently incubated together for 30 m at 37 °C. RF2 and RF3 were prepared similarly. RF1-3 was stored at −20 °C prior to use. The fork regression reaction mixtures contain 5 nM of the indicated DNA substrates, 20 mM Tris (pH 8.0), 2 mM magnesium chloride, 1 mM DTT, 100 μg/ml BSA, 50 mM (for $Sc$Rad5) or 15 mM (for flag-$Kl$Rad5-FL) potassium chloride, 2 mM ATP and $Sc$Rad5 or flag-$Kl$Rad5-FL or

their variants at the indicated concentrations. The reactions were allowed to proceed for 10 m at 30 °C and terminated by a 5-minute incubation at 37 °C in the presence of 0.2% SDS and 0.5 mg/ml proteinase K. Reaction products were resolved with 8% native PAGE carried out with a buffer containing 45 mM Tris-borate (pH 8.0) and 1 mM EDTA at 4 °C. The Cy5 signal in the gel was scanned with a Typhoon FLA 9500 fluorescent image analyzer (GE Healthcare).

**Yeast strains, plasmids, and general manipulations.** Yeast strains used in experiments presented in Figs. 3e–g and 5a–c are derivatives of W1588-4C, a RAD5 derivative of W303 (MATa ade2-1 can1-100 ura3-1 his3-11,15 leu2-3,112 trp1-1 rad5-535)[51]. They are listed in Supplementary Table 2. Two strains per genotype were examined in each experiment. All Rad5 constructs used are integrated alleles expressed from the endogenous promoter at the Rad5 locus. Attaching a tandem affinity purification (TAP) tag to ScRad5 does not affect its function[36]. The standard CRISPR-Cas9 method was used to construct the Sc3RE mutation using a synthesized 750-bp DNA fragment containing the mutation (ThermoFisher Scientific) as a template. The correct transformants were first verified using primers Sc3RE-F and Sc3RE-R (Supplementary Table 3) and then by sequencing of the ScRad5 locus. To clone the ScRad5-Sc3RE ORF into the pOAD yeast two-hybrid vector[52], the mutant ScRad5 gene was PCR-amplified using the Sc3RE mutant genomic DNA as a template and the ScRad5-F and ScRad5-R primers (Supplementary Table 3). The construct was confirmed by sequencing. Yeast two-hybrid and DNA damage sensitivity assays were performed using standard procedures. For the yeast two-hybrid assay, activation domain and DNA-binding-domain plasmids were introduced into reporter strains and cells were grown on synthetic complete medium (SC)-Trp-Leu plates. Positive interactions were assessed by growth after spotting cells onto SC-Trp-Leu-His plates or SC-Trp-Leu-His plates supplemented with 3 mM 3-amino-l,2,4-triazole[53]. For DNA damage sensitivity assays, a tenfold serial dilution of yeast cells was spotted onto plates containing the indicated drugs. Plates were incubated for 3 days before photographed.

To examine the ScRad5 expression level in yeast cells, cells from asynchronous yeast cultures were lysed by bead beating in the presence of 20% trichloroacetic acid (TCA). The pellets were recovered by centrifugation and heated to 95 °C for 5 min in a buffer containing 65 mM Tris-HCl (pH 6.8), 8.5% glycerol, 2% SDS, 5% β-ME, and 0.025% bromophenol blue. ScRad5 was detected by western blot using an anti-TAP antibody (P1291, Sigma-Aldrich, 1:12,000 diluted, Fig. 3g) or an anti-Rad5 antibody (yS-15, Santa Cruz Biotechnology, 1:10,000 diluted, Fig. 5b). Equal loading was accessed by staining the membrane with Ponceau S.

**PCNA ubiquitination detection.** 6x histidine-tagged PCNA from yeast protein extracts was pulled down by Ni-NTA resin, followed by western blotting to detect the ubiquitinated PCNA[36]. When indicated, yeast cells were treated with 0.02% MMS for 2 h. Yeast cell extracts were prepared in 55% TCA and dissolved in buffer A (6 M guanidine HCl, 100 mM sodium phosphate (pH 8.0), 10 mM Tris-HCl (pH 8.0)). After supplementing 0.05% Tween 20 and 14.4 mM imidazole, the cell extract was incubated with Ni-NTA resin for 12 h. The resin was washed twice with buffer A supplemented with 0.05% Tween 20 and 4 times with buffer C (8 M urea, 100 mM sodium phosphate (pH 6.3), 10 mM Tris-HCl (pH 6.3), 0.05% Tween 20). PCNA was eluted with HU buffer (8 M urea, 200 mM Tris-HCl (pH 6.8), 1 mM EDTA, 5% SDS, 0.1% bromophenol blue, 1.5% DTT, 200 mM imidazole). The purified fraction was examined with western blot using antibodies against ubiquitin (P4D1, sc-8017, Santa Cruz Biotechnology, 1:1000 diluted) and PCNA (1:6000 diluted)[54].

**Chromatin fractionation.** Yeast cells grown in log phase were subjected to spheroplasting by supplementing the growth medium with 0.6 M sorbitol, 25 mM Tris-HCl (pH 7.5), 10 mM DTT, and purified lytic β-1,3-glucanase. Spheroplasts were washed with a buffer containing 0.4 M sorbitol, 20 mM PIPES–KOH (pH 6.6), 150 mM potassium acetate, 2 mM magnesium acetate, 1× protease inhibitors (Sigma), and lysed by a 5-min incubation on ice in an extraction buffer containing 20 mM PIPES–KOH (pH 6.6), 150 mM potassium acetate, 2 mM magnesium acetate, 1 mM sodium fluoride, 0.5 mM Na₃VO₄, 1× protease inhibitors (Sigma) and 1% Triton X-100. Chromatin sedimentation was carried out with 15-min centrifugation at $16,000 \times g$ on a sucrose cushion (the extraction buffer supplemented with 30% sucrose). The pellets were washed and resuspended with the extraction buffer. Equal amounts of the whole-cell extract, non-chromatin, and chromatin fractions were analyzed with western blot with the anti-Pgk1 (22C5D8, Invitrogen, 1:8000 diluted), anti-histone H3 (ab46765, Abcam, 1:2000 diluted), and anti-TAP (P1291, Sigma-Aldrich, 1:12,000 diluted) antibodies.

**Reporting summary.** Further information on research design is available in the Nature Research Reporting Summary linked to this article.

## Data availability

The structure factors and coordinates for the native and mercury-derivatized KlRad5 crystals have been deposited into the Protein Data Bank (http://www.rcsb.org), with the accession codes 6L8N and 6L8O, respectively. The source data underlying Figs. 2d–e, 3a–c, 3f–g, 4a–b, 4d–e and 5a–b and Supplementary Figs. 5d, 6b, 7a–b, 7d, 7f, 8a–e, 9a–e, 10a–b and 11d–i are provided as a Source Data file. The previously published protein structures were retrieved from Protein Data Bank, URLs to these structures are provided in the related figure legends. Other data are available from the corresponding author upon reasonable request.

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

## Acknowledgements

We thank scientists at the Shanghai Synchrotron Radiation Facility beamline BL17U1 for setting up the beamline and assistance during diffraction data collection, the molecular biology core facility at Institute of Biochemistry and Cell Biology, Chinese Academy of Sciences for assistance with FP and dynamic light scattering experiments, Prof. Lu Zhu at Tianjin Medical University for assistance with FP experiments, the Protein Chemistry Facility at the Center for Biomedical Analysis of Tsinghua University for assistance with hydrogen-deuterium exchange experiments, Prof. Helle Ulrich at the Institute of Molecular Biology, Mainz for the kind gift of the *Sc*Rad5 expression plasmid, Dr. Nancy Hollingsworth at Stony Brook University for the kind gift of the yeast strain 334, Dr. Bruce Stillman at Cold Spring Harbor Laboratory for the kind gift of the anti-PCNA antibody, Dr. Huilin Zhou at University of California San Diego for the kind gift of the lytic β-1,3-glucanase expression plasmid, Prof. Liang Tong at Columbia University for comments that greatly improved the manuscript. This work is supported by Natural Science Foundation of China (general grants 31870769 and 32071205 to S.X.) and the National Institute of General Medical Sciences (grants GM080670 and GM131058 to X.Z.).

## Author contributions

M.S. performed the crystallographic studies and the ATPase, FP, *Kl*Rad5/HLTF-DNA co-precipitation, ubiquitin-chain extension, and EMSA experiments, with assistance from X.G. and X.L. N.D. and X.Z. performed the yeast two-hybrid and in vivo experiments. Q.W. and H.N. performed the fork regression and *Sc*Rad5-DNA co-precipitation experiments. C.C., H.Z., and Y.H. performed the negative staining EM experiments. S.Z., X.T., and H.D. performed the hydrogen-deuterium exchange experiments. J.Y. and M.S. performed the Rad5-PCNA co-precipitation experiments. X.X., L.Z., and M.X. performed the initial optimization of Rad5 expression, purification, and crystallization. Y.G. provided assistance throughout the project. X.Z. made conceptual contributions to the project. X.Z., H.N., and S.X. wrote the paper. S.X. directed and supervised the research.

## Competing interests

The authors declare no competing interests.
