## [Peer Review File · Nature Communications]

Reviewers' comments:

Reviewer #1 (Remarks to the Author):

In this manuscript, the authors report on the structure of an almost full length Rad5 construct of *K. lactis*. Rad5 is an ATPase and ubiquitin ligase (E3) with an important role in DNA damage bypass. Due to its multifunctionality, a high-resolution structure is valuable as a basis for understanding how the different activities of Rad5 may be coordinated, and why they reside in a single polypeptide.

In the structure presented here, the HIRAN-, SNF2- and RING-domain of Rad5 are well resolved and resemble previously solved structures of these domains. Compared to SNF2-domains in other proteins, the KIRad5 SNF2-domain rests in an ATPase-inactive conformation in the absence of DNA, as the two halves of the active site are far apart. The authors therefore conclude that Rad5 undergoes a major structural rearrangement upon activation (most likely induced by binding to dsDNA). Furthermore, they identified a conserved positively charged motif in the HIRAN domain that is involved in PCNA and ssDNA binding, PCNA poly-ubiquitylation and replication fork regression. In vivo, the corresponding mutation (3RE) causes sensitivity to the DNA-damaging agent MMS and the replication stress inducer hydroxyurea and a failed recruitment of Rad5 to damaged chromatin. Additionally, the authors report that the HIRAN domain of KIRad5, in contrast to that of the human Rad5 homolog HLTF, does not have a preference for free 3'-OH ends of ssDNA, suggesting two distinct ssDNA-binding modes and, as a consequence, two sub-classes of Rad5-like proteins.

Overall, this is a predominantly structural study. The authors provide support for a series of published experimental observations (mainly on budding yeast Rad5) with structural information that largely confirms the expectations based on the previous genetic and biochemical work. The structure itself does not hold a big surprise, but will likely be useful for understanding how the different functionalities of Rad5 are coordinated. Unfortunately, the authors have not taken their study to this level. They follow a number of phenotypes and use the HIRAN domain mutant to probe into the contribution of this domain to Rad5 activities, but their results are largely confirmatory.

Major comments:

1. It would have been interesting if the authors had substantiated some of the novel characteristics of Rad5 that newly emerge from the structure, such as the conformational change upon DNA binding or the "mediator helix", with selected site-directed mutagenesis, followed by phenotypic analysis.
2. The authors make a big point about the differences between Rad5 and HLTF, how the HIRAN domains define two subclasses of Rad5 homologs. Yet, in their structure critical non-conserved residues are not visible. They thus base their conclusions on experiments done with Rad5. In order to support their claims about the different binding modes, I think it would be appropriate to conduct these experiments with Rad5 and HLTF in a side-by-side comparison. If this is not done, they cannot exclude differences in the experimental set-up as a cause of the differences between the published HLTF data and their own observations.
3. It is unclear why the authors use *S. cerevisiae* Rad5 and not (also) KIRad5 for the replication fork regression assays. In this manner, it remains unclear whether the classification of the Rad5 family into two branches is actually valid. Moreover, the purification from budding yeast increases the possibility of co-purifying Rad5-associated proteins with a function in replication fork regression. This is especially true for the comparison of Rad5 WT and 3RE, because the 3RE is not interacting with PCNA anymore.
4. One of the most interesting features of Rad5 appears to be the HIRAN domain, involved in binding to both ssDNA and the substrate, PCNA. Hence, this domain and its mutants would be crucial in an analysis on how chromatin association or DNA binding relates to E3 activity. Is Rad5's E3 activity stimulated by the HIRAN domain binding to ssDNA? Or are the interactions mutually

exclusive? It is a pity that these interesting points are not followed up at all, as this type of question would really provide answers about how Rad5 functions.

5. Most of the observed defects of the 3RE mutant can be explained by the loss of the interaction with PCNA due to a non-functional HIRAN domain. This was already shown by Fan and colleagues. The HIRAN domain of Rad5 is required for the interaction with PCNA (Fan et al., Current genetics 2018).

6. The authors would need to discuss the relevance of fork regression for *S. cerevisiae*. The activity is well established in human systems as a response to replication stress, but this is not true for yeast. Thus, the different propensities to undergo fork regression may or may not relate to the distinct ssDNA-binding properties of Rad5 versus HLTf. This should be considered in more detail.

Minor comments:

7. Typos/English:

Page 3: SHRPH  SHPRH

Page 4: "We screened through a number of fungal species and WERE were able to CRYSTALLIZE in the absence of DNA a nearly full-length fragment of Rad5 ..."

Page 8: methansulfonate (MMS)  methyl methansulfonate

8. Methods: The genetic backgrounds of the yeast strains used in Figure 3f and Figure 5 need to be described. How are the Rad5 constructs expressed? It would be important to show the expression levels of the different Rad5 mutants in budding yeast.

9. Figures 3c: It is not clear, which detection method was used.

Reviewer #2 (Remarks to the Author):

The manuscript described the crystal structure of nearly full-length Rad5 from yeast *K. lactis*. Rad5 has multiple domains responsible for ubiquitin ligase and fork regression activities that are crucially involved in DNA damage tolerance. Up to now, the DNA binding domain of HLTf, a human homolog of Rad5, is structurally characterized, whereas structural insight of whole structure of Rad5 has been missing. This study firstly revealed the overall structure of Rad5. The validation reports showed that the structures were well-refined. The Snf2 domain adopts ATPase-inactive resting state in the absence of DNA. That consideration is well supported by experimental evidences. Although conformational change is so large between ATP-inactive and ATP-competent state in KI-Rad5, the mechanism of activation is basically conserved with some Snf2 proteins as mentioned by the author.

The reviewer considers that this study provides new significant findings as follows. First, the HIRAN of KI-Rad5 (KI-HIRAN) interacts with ssDNA by a mechanism distinct from that of human HLTf (Hs-HIRAN). Second, KI-HIRAN interacts with PCNA and is crucially involved in ubiquitin chain extension on PCNA. Third, three positive residues (R190/R228/R240) are crucial for these functions. Although the manuscript is concise and well organized, the reviewer has several concerns or comments about these findings, as follows.

1) The authors described that the DNA binding mechanism of the KI-HIRAN is different from that of HLTf. This consideration would be supported by amino acid sequence alignment. The residues Hs-HIRAN crucial for DNA binding are not conserved with KI-HIRAN. To deeply characterize the KI-HIRAN, binding assays of the KI-HIRAN domain only (WT and 3RE) using structurally different DNAs, ssDNA, blunt-ended dsDNA, 3'-overhanged dsDNA, and 5'-overhanged dsDNA. Affinity for DNA with different structures would be significant to understand the function of KI-HIRAN. The affinity may be associated with interpretation of fork regression activity for different forks. Also, it is more informative if the authors could propose a model for the interaction of HIRAN with DNA or deeply discuss the interaction.

2) The authors show Kl-Rad5 interacts with PCNA through the conserved three positive residues. Functional difference of the HIRAN of Kl-Rad5 from that of HlTF is a major finding in this study. It is significant that the interaction is specific in yeast or not. So, it is required to demonstrate the interaction analysis between HIRAN of HlTF and PCNA. The authors described that the acidic region of yeast PCNA might be involved in the interaction with the HIRAN. To confirm it, interaction analysis using PCNA mutant is also required.

Recent study has reported that HlTF has a PCNA-interacting motif, APIM, in the C-terminal region. As shown in Supplementary Fig.1, the motif would be conserved. It will be more informative if the authors could evaluate or discuss the interaction with PCNA through the motif.

3) The authors describe that the conserved positive residues are involved in multiple activities of Rad5, and these residues are crucial for both interactions with PCNA and DNA. That implies competitive interaction with PCNA and DNA. In physiological environment, PCNA might be stalled at the primer-template junction. If so, the competitive interaction might be unfavorable for extension of ubiquitin chain and fork regression. The authors should discuss that issue.

4) Related to above comment, the reviewer has interest in extension of ubiquitin chain on PCNA in the presence of DNA and comparison with the previous study (Masuda et al., NAR, 2018).

Minor points:

1) In the abstract and the introduction sections, the authors said 3.3 angstrom is high-resolution. It is too strong. Simply, "the crystal structure" could be better.

2) R610 should be labelled in the right panel of Fig. 2C.

3) It is difficult to understand expected domain movement in Supplementary Fig. 5b. The reviewer recommends to reproduce and enlarge it. In addition, the reviewer also recommends that E628, Q1063, and modeled ATP are shown in the figure.

4) Page 3, line 56; "translesion DNA polymerases"
"translesion DNA polymerase, Rev1," would be better.

Reviewer #3 (Remarks to the Author):

I have been asked to assess the HDX-MS work in the manuscript entitled "Structural basis for the multi-activity factor Rad5 in replication stress tolerance" submitted for publication in Nature Communications.

The results of the HDX-MS work is commented in lines 125-128 and 141-143 of the main manuscript and the experiments explained in lines 495-511 of the methods section. Data are presented in Supplementary Figure 5d and 5e and Supplementary Table 1.

Major concerns:

1. Although it is stated in the methods section that hydrogen-deuterium exchange experiments were conducted in the presence and absence of dsDNA, only the deuterium uptake with dsDNA is reported in Supplementary Table 1. The numbers in the "Free K/Rad5" column are theoretical max deuterium contents, neglecting back-exchange. They basically correspond to the number of backbone amides in the peptide besides proline. The difference calculated between this theoretical maximum deuterium uptake and the deuterium uptake measured in the presence of dsDNA has no meaning. It is the difference between the experimentally observed deuterium content with and without dsDNA which would be relevant.

2. The deuterium content measured in the presence of dsDNA are for most peptide suspiciously low, which may reflect an experimental problem. I know from own experience that HDXMS on a protein/RNA complex is very tricky. Quenching of the hydrogen deuterium exchange reaction involves lowering the pH to a point where the protein will be highly positively charged and the phosphodiester backbone of the DNA still negatively charged. The result is an electrostatic complex which may prove very difficult to digest with pepsin without denaturant/Chaotrope. Undigested Protein/DNA complexes will likely carry-over in the LC system, lose all its deuterium to back-exchange and slowly be degraded by new shots of pepsin being injected with the samples and eluted into the MS. That way what is being measured is not the deuterium content of the

injected sample but the much lower deuterium content of peptides slowly released from the carry-over protein on the LC-system. I cannot say if that is what is happening in this case, but the measured deuterium contents are SO LOW that it would be my best guess.

Conclusion:

Based on my major concerns outlined above I cannot recommend publishing these HDXMS data.

We appreciate the insightful comments from the reviewers. Our responses to the points they raised are listed below in red.

Reviewer #1 (Remarks to the Author):

In this manuscript, the authors report on the structure of an almost full length Rad5 construct of *K. lactis*. Rad5 is an ATPase and ubiquitin ligase (E3) with an important role in DNA damage bypass. Due to its multifunctionality, a high-resolution structure is valuable as a basis for understanding how the different activities of Rad5 may be coordinated, and why they reside in a single polypeptide.

In the structure presented here, the HIRAN-, SNF2- and RING-domain of Rad5 are well resolved and resemble previously solved structures of these domains. Compared to SNF2-domains in other proteins, the KIRad5 SNF2-domain rests in an ATPase-inactive conformation in the absence of DNA, as the two halves of the active site are far apart. The authors therefore conclude that Rad5 undergoes a major structural rearrangement upon activation (most likely induced by binding to dsDNA). Furthermore, they identified a conserved positively charged motif in the HIRAN domain that is involved in PCNA and ssDNA binding, PCNA poly-ubiquitylation and replication fork regression. In vivo, the corresponding mutation (3RE) causes sensitivity to the DNA-damaging agent MMS and the replication stress inducer hydroxyurea and a failed recruitment of Rad5 to damaged chromatin. Additionally, the authors report that the HIRAN domain of KIRad5, in contrast to that of the human Rad5 homolog HLTF, does not have a preference for free 3'-OH ends of ssDNA, suggesting two distinct ssDNA-binding modes and, as a consequence, two sub-classes of Rad5-like proteins.

Overall, this is a predominantly structural study. The authors provide support for a series of published experimental observations (mainly on budding yeast Rad5) with structural information that largely confirms the expectations based on the previous genetic and biochemical work. The structure itself does not hold a big surprise, but will likely be useful for understanding how the different functionalities of Rad5 are coordinated. Unfortunately, the authors have not taken their study to this level. They follow a number of phenotypes and use the HIRAN domain mutant to probe into the contribution of this domain to Rad5 activities, but their results are largely confirmatory.

We greatly appreciate the constructive comments from this reviewer. We addressed her/his comments thoroughly with new data and better writing as detailed below.

Major comments:

1. It would have been interesting if the authors had substantiated some of the novel characteristics of Rad5 that newly emerge from the structure, such as the conformational change upon DNA binding or the "mediator helix", with selected site-directed mutagenesis, followed by phenotypic analysis.

In figure 2 we have shown that mutations disrupting dsDNA binding to KIRad5 and the interaction stabilizing the ATPase-inactive form of KIRad5 inhibit and stimulate its ATPase activity, respectively, which is consistent with our model that dsDNA induces large conformational changes in Rad5 to activate its ATPase activity. In the revised manuscript, we have included new experiments to characterize the function of the mediator helix. We probed how disrupting the Snf2-HIRAN interaction centering around the mediator helix affects Rad5's

activities. Briefly, we introduced amino acid substitutions in the mediator helix that form extensive interactions with the Snf2 and HIRAN domains and deleted the β 1- β 3 insertion in the Snf2 domain lobe 1, which forms extensive interactions with the mediator helix and the HIRAN domain. We found that both the point substitutions and the β 1- β 3 deletion severely inhibited the ubiquitin ligase and ATPase activities of *K/Rad5*, suggesting that the Snf2-HIRAN interaction is critical for two main activities of Rad5. These new data are presented in the new supplementary figures 4d-e. They support a model that the Snf2-HIRAN interaction centering around the mediator helix is essential for stabilizing the overall structure of Rad5 and its multiple activities.

2. The authors make a big point about the differences between Rad5 and HLTF, how the HIRAN domains define two subclasses of Rad5 homologs. Yet, in their structure critical non-conserved residues are not visible. They thus base their conclusions on experiments done with Rad5. In order to support their claims about the different binding modes, I think it would be appropriate to conduct these experiments with Rad5 and HLTF in a side-by-side comparison. If this is not done, they cannot exclude differences in the experimental set-up as a cause of the differences between the published HLTF data and their own observations.

We have performed the suggested side-by-side experiments. Our co-precipitation experiments indicated that *K/Rad5*'s HIRAN domain mediate interactions with PCNA (Fig. 3a). We have performed similar co-precipitation experiments with the human PCNA and HLTF or its HIRAN domain (new supplementary Fig. 7e), which suggested that the human PCNA and HLTF do not interact. We have also performed ssDNA co-precipitation experiments on HLTF and Rad5 side-by-side (new Supplementary Fig. 8a), which confirmed the reported requirement of the free ssDNA 3'-hydroxyl group for HLTF binding¹ and indicated that it is not required for Rad5 binding. Together, these data indicate that the HIRAN domains in Rad5 and HLTF have distinct functions.

3. It is unclear why the authors use *S. cerevisiae* Rad5 and not (also) *K/Rad5* for the replication fork regression assays. In this manner, it remains unclear whether the classification of the Rad5 family into two branches is actually valid. Moreover, the purification from budding yeast increases the possibility of co-purifying Rad5-associated proteins with a function in replication fork regression. This is especially true for the comparison of Rad5 WT and 3RE, because the 3RE is not interacting with PCNA anymore.

The replication fork regression activity of *S. cerevisiae* Rad5 (*ScRad5*) is well established in the literature²⁻⁴, thus provides a nice setup for testing this activity and allows comparisons between our findings and the literature. We have included this rationale in the revised text. We believe that the replication fork regression activity we observed is not due co-purified protein(s) for the following reasons. First, we followed the same purification protocol used by other groups that reported replication fork regression for *ScRad5* and purified *ScRad5* close to homogeneity. Second, a previous study has shown that *ScRad5* with the D681A/E682A substitution also purified from yeast does not possess the fork regression activity², supporting that the observed replication fork regression activity is not due to factors co-purified with *ScRad5* from yeast. Third, *ScRad5* interacting proteins have been extensively examined by several groups and none of its interactors showed replication fork regression activity.

We note that the *ScRad5*-catalyzed replication fork regression does require PCNA, and our replication fork regression assay does not involve PCNA, following published protocols. Thus, the lack of replication fork regression activity observed for the *Sc3RE*-substituted *ScRad5* reflects a defect independent of PCNA.

4. One of the most interesting features of Rad5 appears to be the HIRAN domain, involved in binding to both ssDNA and the substrate, PCNA. Hence, this domain and its mutants would be crucial in an analysis on how chromatin association or DNA binding relates to E3 activity. Is Rad5's E3 activity stimulated by the HIRAN domain binding to ssDNA? Or are the interactions mutually exclusive? It is a pity that these interesting points are not followed up at all, as this type of question would really provide answers about how Rad5 functions.

The reviewer's points are well taken, and we have performed the suggested experiments. Our new data show that ssDNA strongly inhibits the *K/Rad5*-catalyzed PCNA-anchored ubiquitin-chain extension (new Supplementary Figs 9a-b). In contrast, it does not inhibit the *K/Rad5*-stimulated free ubiquitin-chain extension by Ubc13-Mms2 (new Supplementary Figs 9c-d), which suggests that ssDNA does not impede its ubiquitin ligase activity per se. In fact, consistent with previous studies on HLTF^{5,6}, ssDNA was found to stimulate this activity. Together, these data are consistent with a model that ssDNA competes with PCNA for binding to Rad5's HIRAN domain and prevents its recruitment for ubiquitination but does not impede its ubiquitin ligase activity.

We went further to test the effect of dsDNA. Our data suggested that the ssDNA binding site in *K/Rad5*'s HIRAN domain also mediate interactions with dsDNA (new Supplementary Figs 8b-c). Consistent with this data, we found similar effects of dsDNA on the *K/Rad5*-catalyzed PCNA-anchored and free ubiquitin-chain extension (Supplementary Fig. 9a-d). Together, these new data suggest that both dsDNA and ssDNA inhibit the Rad5-catalyzed PCNA ubiquitination but not its ubiquitin ligase activity. Moreover, ssDNA and dsDNA further suppress the severely inhibited PCNA poly-ubiquitination by the 3RE or the Δ HIRAN variants of *K/Rad5*, suggesting that regions outside of Rad5's HIRAN domain also contribute to DNA's modulation on its PCNA ubiquitination reaction.

Future *in vivo* studies are required to validate our hypotheses regarding DNA's modulation on the Rad5-catalyzed PCNA ubiquitination and probe its physiological function. In the experiments described above, we used the widely accepted ubiquitin-PCNA fusion protein substrate, which is not loaded to DNA⁷. Whereas inside the cell, PCNA is loaded to DNA during DNA replication. We have added a third paragraph in the "Discussion" section to discuss our data related to the possible modulation of DNA on the Rad5-catalyzed PCNA ubiquitination in cells.

5. Most of the observed defects of the 3RE mutant can be explained by the loss of the interaction with PCNA due to a non-functional HIRAN domain. This was already shown by Fan and colleagues. The HIRAN domain of Rad5 is required for the interaction with PCNA (Fan et al., Current genetics 2018).

The work by Fan et al reported yeast two-hybrid data that suggests interaction between Rad5's HIRAN domain and PCNA⁸. Interactions suggested by yeast two-hybrid assays could be mediate by other proteins and caused by indirect effects. Our data demonstrate for the first time that Rad5's HIRAN directly binds to PCNA. Importantly, our data suggest that defects caused by the 3RE substitution are not only related to defects in PCNA binding but also to defects in other activities of Rad5 including DNA binding and replication fork regression. We have revised the text to acknowledge the previous study and highlight our contributions.

6. The authors would need to discuss the relevance of fork regression for *S. cerevisiae*. The activity is well established in human systems as a response to replication stress, but this is not true for yeast. Thus, the different propensities to undergo fork regression may or may not relate

to the distinct ssDNA-binding properties of Rad5 versus HLTF. This should be considered in more detail.

While we agree with the reviewer's point, it is important to note that fork regression occurs in yeast cells, though the frequency of it is much lower than in higher eukaryotic cells. The reason behind this difference is currently unclear, but can be related to several factors, such as that mammalian cells contain more enzymes catalyzing fork regression, have a larger number of forks, and higher frequency of replication stalling. Regardless of these differences, several studies have shown that fork regression by Rad5 is important to cope with replication stress in yeast. Our study shows that although Rad5 and HLTF have distinct ssDNA-binding properties regarding a requirement for the 3'-hydroxyl group, they both catalyze fork regression *in vitro*. We do not mean to relate the difference in ssDNA binding of Rad5 and HLTF to the difference in fork regression frequency. We have revised the first paragraph in the "Discussion" section to include a discussion on the relevance of the Rad5-catalyzed fork regression in yeast.

Minor comments:

7. Typos/English:

Page 3: SHRPB  SHPRH

Page 4: "We screened through a number of fungal species and WERE were able to CRYSTALLIZE in the absence of DNA a nearly full-length fragment of Rad5 ..."

Page 8: methansulfonate (MMS)  methyl methansulfonate

We thank the reviewer for pointing out these typos and have corrected them.

8. Methods: The genetic backgrounds of the yeast strains used in Figure 3f and Figure 5 need to be described. How are the Rad5 constructs expressed? It would be important to show the expression levels of the different Rad5 mutants in budding yeast.

The genetic background of the yeast strains (W303) is described in the "Yeast Strains, plasmids, and general manipulations" section in Methods. All Rad5 constructs used are integrated alleles expressed from the endogenous promoter at the Rad5 locus. We have previously shown that neither the D681A/E682A substitution nor the Q1061A substitution affects the ScRad5 protein level⁹. We have included new data to show that the I916A substitution doesn't do so either (new Fig. 5c) and the Sc3RE substitution moderately reduced the ScRad5 protein level (new Fig. 3g). Although the Sc3RE substitution reduced the ScRad5 protein level, it is unlikely that the moderate reduction alone can account for the severely inhibited PCNA ubiquitination (Fig. 3f) and ScRad5-chromatin association (Fig. 5a) or the significantly increased sensitivity towards MMS or HU (Fig. 5b). These severe defects suggest that the Sc3RE substitution impairs multiple activities of ScRad5.

9. Figures 3c: It is not clear, which detection method was used.

This figure shows SDS PAGE analysis of the ubiquitin-chain extension reaction. The amount of the products (Ub₂-Ub₆, which indicates ubiquitin-chains with 2-6 ubiquitin moieties) are used as a measure of Rad5's ubiquitin ligase activity. We have revised the figure legend to make it clearer.

Reviewer #2 (Remarks to the Author):

The manuscript described the crystal structure of nearly full-length Rad5 from yeast *K. lactis*. Rad5 has multiple domains responsible for ubiquitin ligase and fork regression activities that are crucially involved in DNA damage tolerance. Up to now, the DNA binding domain of HLTF, a human homolog of Rad5, is structurally characterized, whereas structural insight of whole structure of Rad5 has been missing. This study firstly revealed the overall structure of Rad5. The validation reports showed that the structures were well-refined. The Snf2 domain adopts ATPase-inactive resting state in the absence of DNA. That consideration is well supported by experimental evidences. Although conformational change is so large between ATP-inactive and ATP-competent state in KI-Rad5, the mechanism of activation is basically conserved with some Snf2 proteins as mentioned by the author.

The reviewer considers that this study provides new significant findings as follows. First, the HIRAN of KI-Rad5 (KI-HIRAN) interacts with ssDNA by a mechanism distinct from that of human HLTF (Hs-HIRAN). Second, KI-HIRAN interacts with PCNA and is crucially involved in ubiquitin chain extension on PCNA. Third, three positive residues (R190/R228/R240) are crucial for these functions. Although the manuscript is concise and well organized, the reviewer has several concerns or comments about these findings, as follows.

We greatly appreciate the reviewer's positive comments and constructive suggestions. We address her/his comments thoroughly with new data and better writing as detailed below.

1) The authors described that the DNA binding mechanism of the KI-HIRAN is different from that of HLTF. This consideration would be supported by amino acid sequence alignment. The residues Hs-HIRAN crucial for DNA binding are not conserved with KI-HIRAN. To deeply characterize the KI-HIRAN, binding assays of the KI-HIRAN domain only (WT and 3RE) using structurally different DNAs, ssDNA, blunt-ended dsDNA, 3'-overhanged dsDNA, and 5'-overhanged dsDNA. Affinity for DNA with different structures would be significant to understand the function of KI-HIRAN. The affinity may be associated with interpretation of fork regression activity for different forks. Also, it is more informative if the authors could propose a model for the interaction of HIRAN with DNA or deeply discuss the interaction.

We have performed the suggested experiments with the wild type *K/HIRAN*. We purified *K/HIRAN* via an engineered chitin-binding-domain/intein tag followed by intein-catalyzed tag cleavage. FP experiments using the purified *K/HIRAN* indicate that it binds both ssDNA and dsDNA and its affinity towards the former is significantly stronger. It also binds dsDNA with overhanging ssDNA regions and the affinity is comparable to the affinity towards ssDNA (new Supplementary Fig. 8b).

We were not able to purify the 3RE substituted *K/HIRAN*. The protein appears to be misfolded since it precipitated after tag cleavage, although the 3RE-substituted *K/Rad5* is well-folded judging from our size exclusion chromatography (see Fig. R1 attached). We attempted to increase protein solubility by changing the tag to maltose binding protein (MBP) but found that MBP-tagged *K/HIRAN* was degraded during expression.

Proposing a model for the interaction between Rad5's HIRAN domain and DNA and its function in the Rad5-catalyzed replication fork regression is difficult based on the current data. We are planning future studies to elucidate the mechanism of the interaction between Rad5's HIRAN domain and DNA and provide insights into its function.

2) The authors show KI-Rad5 interacts with PCNA through the conserved three positive residues. Functional difference of the HIRAN of KI-Rad5 from that of HLTF is a major finding

in this study. It is significant that the interaction is specific in yeast or not. So, it is required to demonstrate the interaction analysis between HIRAN of HLTF and PCNA. The authors described that the acidic region of yeast PCNA might be involved in the interaction with the HIRAN. To confirm it, interaction analysis using PCNA mutant is also required. Recent study has reported that HLTF has a PCNA-interacting motif, APIM, in the C-terminal region. As shown in Supplementary Fig.1, the motif would be conserved. It will be more informative if the authors could evaluate or discuss the interaction with PCNA through the motif.

We performed the suggested interaction experiments and found that HLTF's HIRAN domain does not possess detectable affinity towards PCNA (new Supplementary Fig. 7e). Our new data on the *K/Rad5*-PCNA interaction with PCNA mutants suggest that a cleft between monomers in the trimeric PCNA outer surface mediates interactions with Rad5 (new Supplementary Figs 7c-d).

The APIM motif is composed of five consensus residues (K/R)-(F/Y/W)-(L/I/V/A)-(L/I/V/A)-(K/R). A recent study suggested that an APIM motif in HLTF (residues 959-963) mediates interactions with PCNA, but a direction interaction was not detected⁵. Our structure shows that the equivalent region in *K/Rad5* (residues 1069-1073, RFIID), especially F1070, I1071 and I1072, are largely buried in the interior of Snf2 domain lobe 2 (see Fig. R2 attached). Unfolding of the Snf2 domain lobe 2 is required to expose these residues to interact with PCNA. Therefore, it is unlikely that the putative APIM motif in Rad5 or HLTF mediates interactions with PCNA. We have revised the third paragraph in the "The HIRAN domain is critical for the Rad5-catalyzed PCNA-anchored ubiquitin-chain extension" section to include a description of the putative APIM motif.

3) The authors describe that the conserved positive residues are involved in multiple activities of Rad5, and these residues are crucial for both interactions with PCNA and DNA. That implies competitive interaction with PCNA and DNA. In physiological environment, PCNA might be stalled at the primer-template junction. If so, the competitive interaction might be unfavorable for extension of ubiquitin chain and fork regression. The authors should discuss that issue.

This is largely related to point 4 of reviewer 1, please refer to our response to that point for details. In brief, we found an inhibitory effect of DNA on the Rad5-catalyzed PCNA poly-ubiquitination but not Rad5's ubiquitin ligase activity per se, consistent with a model that DNA competes with PCNA for binding to Rad5 to inhibit its ubiquitination. Future *in vivo* experiments are required to validate this model and probe the physiological function of DNA's modulation on the Rad5-catalyzed PCNA ubiquitination. Likewise, additional experiments are required to probe the possible modulation of PCNA on the HIRAN-DNA interaction at stalled replication forks, which may affect Rad5's recruitment to these locations and/or the Rad5-catalyzed fork regression. We have added a third paragraph in the "Discussion" section to include the above discussions.

4) Related to above comment, the reviewer has interest in extension of ubiquitin chain on PCNA in the presence of DNA and comparison with the previous study (Masuda et al., NAR, 2018).

In the referenced paper⁵, the authors performed enzymatic studies of HLTF in catalyzing PCNA poly-ubiquitination and tested how various factors influence the reaction. They concluded that RFC and DNA influence this reaction *in vitro*. As detailed in our response to point 4 of reviewer 1, our new experiments suggest inhibitory effects of both ssDNA and

dsDNA towards the Rad5-catalyzed ubiquitin-chain extension on PCNA. The ubiquitin-PCNA fusion protein that is not loaded to DNA is used as substrate in these experiments. We plan to test the effect of PCNA loading to DNA and other factors including RFC in a future study.

Minor points:

- 1) In the abstract and the introduction sections, the authors said 3.3 angstrom is high-resolution. It is too strong. Simply, "the crystal structure" could be better.
- 2) R610 should be labelled in the right panel of Fig. 2C.
- 3) It is difficult to understand expected domain movement in Supplementary Fig. 5b. The reviewer recommends to reproduce and enlarge it. In addition, the reviewer also recommends that E628, Q1063, and modeled ATP are shown in the figure.
- 4) Page 3, line 56; "translesion DNA polymerases"
"translesion DNA polymerase, Rev1," would be better.

We thank the reviewer for pointing these out and have revised the manuscript according to the reviewer's suggestions.

Reviewer #3 (Remarks to the Author):

I have been asked to assess the HDX-MS work in the manuscript entitled "Structural basis for the multi-activity factor Rad5 in replication stress tolerance" submitted for publication in Nature Communications.

The results of the HDX-MS work is commented in lines 125-128 and 141-143 of the main manuscript and the experiments explained in lines 495-511 of the methods section. Data are presented in Supplementary Figure 5d and 5e and Supplementary Table 1.

Major concerns:

1. Although it is stated in the methods section that hydrogen-deuterium exchange experiments were conducted in the presence and absence of dsDNA, only the deuterium uptake with dsDNA is reported in Supplementary Table 1. The numbers in the "Free K/Rad5" column are theoretical max deuterium contents, neglecting back-exchange. They basically correspond to the number of backbone amides in the peptide besides proline. The difference calculated between this theoretical maximum deuterium uptake and the deuterium uptake measured in the presence of dsDNA has no meaning. It is the difference between the experimentally observed deuterium content with and without dsDNA which would be relevant.

2. The deuterium content measured in the presence of dsDNA are for most peptide suspiciously low, which may reflect an experimental problem. I know from own experience that HDXMS on a protein/RNA complex is very tricky. Quenching of the hydrogen deuterium exchange reaction involves lowering the pH to a point where the protein will be highly positively charged and the phosphodiester backbone of the DNA still negatively charged. The result is an electrostatic complex which may prove very difficult to digest with pepsin without denaturant/Chaotrope. Undigested Protein/DNA complexes will likely carry-over in the LC system, lose all its deuterium to back-exchange and slowly be degraded by new shots of pepsin being injected with the samples and eluted into the MS. That way what is being measured is not the deuterium content of the injected sample but the much lower deuterium content of peptides slowly released from the carry-over protein on the LC-system. I cannot say

if that is what is happening in this case, but the measured deuterium contents are SO LOW that it would be my best guess.

Conclusion:

Based on my major concerns outlined above I cannot recommend publishing these HDXMS data.

We greatly appreciate the constructive comments of this reviewer and have performed additional HDXMS experiments and revised the manuscript to address the points he/she raised. We have replaced the quenching buffer in our new HDXMS experiments to include a chaotropic agent (4M guanidinium hydrochloride), a reducing agent (500 mM tris(2-carboxyethyl)phosphine) and 200 mM citric acid to facilitate proteolytic digestion, following a published protocol¹⁰. After each LC-MS/MS run, we thoroughly washed away the leftover samples in the LC column. We also carried out three independent repeats of each experiment. We think that both *K/Rad5* and its complex with dsDNA were efficiently digested in our new HDXMS experiments for the following reasons. First, the digested peptides identified by our new HDXMS experiments in the absence or presence of dsDNA covered approximately 90% of the *K/Rad5* sequence. Second, overall the deuterization level of peptides measured in the presence of dsDNA were quite significant and were not significantly lower than the deuterization level measured in the absence of it. Third, the maximum deuterium content of peptides estimated by 24-hour D₂O incubation in the absence or presence of dsDNA were mostly about 50-70% of the theoretical maximum values, which is comparable to numbers reported in the literature. In the attached table R1, the maximum deuterium content of selected peptides are presented. The deuterium content of these peptides in experiments with 60-second or 300-second D₂O incubation with dsDNA is significantly different from the deuterium content in experiments without dsDNA (see supplementary figures 6a-b in the manuscript).

The new HDXMS data are presented in the new supplementary figures 6a-b, which show the deuterization level of the selected peptides mentioned above. The deuterization level is defined as the deuterium content divided by the maximum deuterium content. Upon dsDNA binding, the deuterization level of peptides 548-554, 684-700, 1039-1049 and 1081-1092 decrease, of peptides 771-778 and 1093-1108 increase. These data are consistent with our structural model for the conformational change in *K/Rad5* upon dsDNA binding. We have removed supplementary table 1 from the manuscript, the related data is presented in the source data associated with supplementary figure 6b.

Table R1 Maximum deuterium content of peptides estimated by 24-hour D₂O incubation in the absence or presence of dsDNA*

Peptides	Free K/Rad5 [#]			K/Rad5 with dsDNA [#]		
	Exp 1	Exp 2	Exp 3	Exp 1	Exp 2	Exp 3
188-226	16.614	17.283	17.647	17.115	16.814	16.445
	51.92%	54.01%	55.15%	53.48%	52.54%	51.39%
548-554	1.934	1.737	1.586	1.821	1.628	1.572
	48.35%	43.43%	39.65%	45.53%	40.71%	39.30%
684-700	8.799	8.509	8.486	8.704	8.237	8.092
	62.85%	60.78%	60.62%	62.17%	58.84%	57.80%
739-753	9.038	8.237	7.618	8.203	8.243	8.112
	69.53%	63.36%	58.60%	63.10%	63.41%	62.40%
771-778	3.977	3.952	3.816	3.870	3.784	3.736
	66.28%	65.86%	63.60%	64.51%	63.07%	62.27%
1039-1049	3.609	3.641	3.728	3.576	3.553	3.501
	51.56%	52.01%	53.25%	51.09%	50.76%	50.02%
1081-1092	5.855	5.874	5.928	5.728	5.757	5.782
	58.55%	58.74%	59.28%	57.28%	57.57%	57.82%
1093-1108	8.041	8.001	8.328	7.953	7.792	7.842
	57.44%	57.15%	59.49%	56.81%	55.65%	56.01%

*Peptides with significant differences in the deuterization level between experiments with 60-second or 300-second D₂O incubation with and without dsDNA are presented (see supplementary figures 6a-b in the manuscript).

[#]The deuterium content and its ratio to the theoretical maximum deuterium content (lower line) are presented.

Figure R1 Size exclusion chromatography of the wild type and 3RE-substituted *K/Rad5*.

Figure R2 Location of the putative APIM motif in *K/Rad5*. The putative APIM motif in the Snf2 domain lobe 2 is highlighted in gray. Residues in this motif and surrounding secondary structure elements are indicated.

References

- 1 Kile, A. C. *et al.* HLTf's Ancient HIRAN Domain Binds 3' DNA Ends to Drive Replication Fork Reversal. *Mol Cell* **58**, 1090-1100, doi:10.1016/j.molcel.2015.05.013 (2015).
- 2 Blastyak, A. *et al.* Yeast Rad5 protein required for postreplication repair has a DNA helicase activity specific for replication fork regression. *Mol Cell* **28**, 167-175, doi:S1097-2765(07)00547-3 [pii] 10.1016/j.molcel.2007.07.030 (2007).
- 3 Neelsen, K. J. & Lopes, M. Replication fork reversal in eukaryotes: from dead end to dynamic response. *Nat Rev Mol Cell Biol* **16**, 207-220, doi:10.1038/nrm3935 (2015).
- 4 Meng, X. & Zhao, X. Replication fork regression and its regulation. *FEMS Yeast Res* **17**, doi:10.1093/femsyr/fow110 (2017).
- 5 Masuda, Y. *et al.* Regulation of HLTf-mediated PCNA polyubiquitination by RFC and PCNA monoubiquitination levels determines choice of damage tolerance pathway. *Nucleic Acids Res* **46**, 11340-11356, doi:10.1093/nar/gky943 (2018).
- 6 Masuda, Y. *et al.* En bloc transfer of polyubiquitin chains to PCNA in vitro is mediated by two different human E2-E3 pairs. *Nucleic Acids Res* **40**, 10394-10407, doi:10.1093/nar/gks763 (2012).
- 7 Parker, J. L. & Ulrich, H. D. Mechanistic analysis of PCNA poly-ubiquitylation by the ubiquitin protein ligases Rad18 and Rad5. *EMBO J* **28**, 3657-3666, doi:emboj2009303 [pii] 10.1038/emboj.2009.303 (2009).

- 8 Fan, Q. *et al.* Rad5 coordinates translesion DNA synthesis pathway by recognizing specific DNA structures in *saccharomyces cerevisiae*. *Curr Genet* **64**, 889-899, doi:10.1007/s00294-018-0807-y (2018).
- 9 Choi, K. *et al.* Concerted and differential actions of two enzymatic domains underlie Rad5 contributions to DNA damage tolerance. *Nucleic Acids Res* **43**, 2666-2677, doi:10.1093/nar/gkv004 (2015).
- 10 Wales, T. E., Eggertson, M. J. & Engen, J. R. Considerations in the analysis of hydrogen exchange mass spectrometry data. *Methods Mol Biol* **1007**, 263-288, doi:10.1007/978-1-62703-392-3_11 (2013).

REVIEWER COMMENTS

Reviewer #1 (Remarks to the Author):

Overall, the authors have done a good job in revising the manuscript. They have made extensive modifications to the text, clarifying their language and descriptions in several instances, and they have added a set of new experiments that now provide a deeper mechanistic investigation of the different activities of Rad5. In my view, there are still a few open questions, however, which should be addressed, as they pertain to the major novel finding presented in this study, the unique functionality of the Rad5 HIRAN domain. According to the authors' data, this domain functions in a significantly different manner in the yeast Rad5 proteins compared to mammalian HLTF. They observe differential properties with respect to DNA binding, PCNA interaction and PCNA-directed ubiquitin ligase (E3) function. However, what remains vague is the underlying reason for these differences. They could be caused either by the sequence of the HIRAN domain proper, or alternatively by the arrangement of the domain within the overall protein domain architecture. The authors have all the means at hand to clarify this point, and I strongly believe that they should make this effort. It would simply require repeating DNA and PCNA binding using the beta1-3 mutant as well as mutants in the mediator helix to determine whether the apparently unique connectivity between the HIRAN and Snf2 domains is responsible for the coupling of the HIRAN domain to PCNA binding and the competition with ssDNA.

My other comments are listed according to my original points and the authors' responses:

1. To ensure the overall stability of the tested mutants it would be helpful to include some QC data of the purifications (e.g. SEC profiles). In addition, as mentioned above, conclusions about the contribution of the different Rad5 domains on its activities would require a systematically screening of all Rad5 activities with the respective mutants is required (i.e. including interactions with PCNA, ssDNA and dsDNA).
2. Figures S7D and E are still difficult to understand. The authors should revise the labels and figure captions. Especially, they should clearly indicate the associated tags on the respective proteins and indicate which protein was used as the bait in the co-precipitations.
3. The authors have clarified why they used ScRad5 in the replication fork regression assays. Nevertheless, definitive evidence that would allow conclusions about two subfamilies of Rad5 relatives, it would be required to perform this assay with the KIRad5 protein.
4. The results regarding the ubiquitin ligase activity of Rad5 in the presence or absence of ssDNA and dsDNA appear inconsistent. The inhibitory effect of ssDNA and dsDNA is only observable when the enzymatic activity of Rad5 is restricted to the addition of one ubiquitin to Ub-PCNA ("priming" Ub) (Figure S9B). In figure S9A, with non-restricted K63-chain formation activity on Ub-PCNA, no inhibitory effect is detectable. This seems to indicate that the inhibitory effect is only present for the priming Ub on Ub-PCNA. The authors should discuss and try to resolve this. Regarding the effect of PCNA loading: the study quoted by the authors showed that loading has a stimulatory effect on Rad5's ubiquitin ligase activity towards PCNA. I am not suggesting that the authors repeat their assays with loaded Rad5, as this may go beyond the scope of the manuscript, but they should discuss that their assays trying to tease apart ssDNA binding and PCNA binding/ubiquitylation are likely non-physiological, given that the physiological substrate would be loaded PCNA.
5. The authors have responded well to this point.
6. The authors have responded well to this point. Yet, the vastly different propensities of yeast and human cells to undergo fork regression (in yeast these have been directly observed only in checkpoint mutants, reflecting a very non-physiological situation) might suggest that the distinct properties of the two classes of Rad5 relatives in yeast versus humans could be related to that phenomenon.

7. Some typos remain:

Abstract: "automatous" should probably be "autonomous"

Page 3: "an universal" should be "a universal"

Page 4: "We present here crystal structure of ..." should be "We present here a crystal structure of ..."

Page 7: "fiorescent" should be "fluorescent"

8. The authors have responded well to this point.

9. The detection method (Figure 3c) is still not mentioned (Coomassie staining?).

Additional minor comments:

Figure S5d: The authors should include input samples, loading control and indicate the detection method.

Figure S8a: The authors should describe the rationale of using single-strand binding protein (SSB) in the figure caption or the main text.

Reviewer #2 (Remarks to the Author):

The manuscript was carefully revised by taking into account of all of my concerns and comments. The revised version provides more in-depth insights into functions of KI-Rad5 and highlights the functional difference between Rad5 and HLTf. Several concerns are newly arisen in the revised manuscript.

1) Binding experiments revealed that the positive region in HIRAN of KI-Rad5 interacted with the acidic region of PCNA, where D109 and E113 were crucially involved in the interaction. These residues are located in the cleft between PCNA monomers. Based on the trimer structure of PCNA, these residues might be involved in the trimer assembly. In fact, E113A mutation in ScPCNA reduced trimer stability of ScPCNA (Dieckman et al., DNA Rep., 2013). The author should evaluate the assembly of KI-PCNA with E113K mutation by SEC. The reviewer is afraid that instability of PCNA trimer might affect the interaction.

2) The reviewer agrees that ssDNA binding is a common feature for HIRAN domain in Rad5 family members (page 10, line 222-223). In the revised manuscript, the authors reveals that KI-HIRAN binds to ssDNA and dsDNA with blunt ends or 5'- or 3'- overhanging ssDNA regions. And the affinity to ssDNA and dsDNA with overhanging ssDNA regions are much stronger. In this context, the readers might consider that HIRAN of human HLTf has no affinity for dsDNA with blunt ends or 5'- or 3'- overhanging ssDNA regions. However, these results of KI-HLTf are consistent with those of HIRAN of human HLTf (Hishiki et al. JBC, 2015). To avoid misleading, the author should add description about affinity of HIRAN of HLTf for dsDNAs.

Minor comment:

The labels "kT/e" in the electrostatic potential may be "k_B/e", where "k" and "B" are shown as an italics and a subscript, respectively, because k_B would be Boltzmann constant. The reviewer overlooked that in the first review.

Reviewer #3 (Remarks to the Author):

The authors have redone the HDXMS experiments and they now appear trustworthy. The differences in deuteration level with and without dsDNA bound are rather small however, although

statistically significant. Bigger differences may be obtained with further experimentation with shorter incubation times, lower incubation temperature or through increased fraction of dsDNA-bound KIRAD5 relative to unbound K1RAD5 during labeling with deuterium.

Morten Beck Trelle

We greatly appreciate the insightful comments from the reviewers. Our responses to the points they raised are listed below in red.

Reviewer #1 (Remarks to the Author):

Overall, the authors have done a good job in revising the manuscript. They have made extensive modifications to the text, clarifying their language and descriptions in several instances, and they have added a set of new experiments that now provide a deeper mechanistic investigation of the different activities of Rad5. In my view, there are still a few open questions, however, which should be addressed, as they pertain to the major novel finding presented in this study, the unique functionality of the Rad5 HIRAN domain. According to the authors' data, this domain functions in a significantly different manner in the yeast Rad5 proteins compared to mammalian HLTF. They observe differential properties with respect to DNA binding, PCNA interaction and PCNA-directed ubiquitin ligase (E3) function. However, what remains vague is the underlying reason for these differences. They could be caused either by the sequence of the HIRAN domain proper, or alternatively by the arrangement of the domain within the overall protein domain architecture. The authors have all the means at hand to clarify this point, and I strongly believe that they should make this effort. It would simply require repeating DNA and PCNA binding using the beta1-3 mutant as well as mutants in the mediator helix to determine whether the apparently unique connectivity between the HIRAN and Snf2 domains is responsible for the coupling of the HIRAN domain to PCNA binding and the competition with ssDNA.

We performed the suggested experiments. Our new data show that substitutions or the $\Delta\beta 1-3$ truncation that disrupt the HIRAN-Snf2 interaction inhibited multiple activities of *K/Rad5*, including its ATPase activity, binding to dsDNA, ssDNA and PCNA, its ubiquitin ligase activity and the PCNA ubiquitination reaction it catalyzes. These new data are presented in the supplementary figure 11 (supplementary figure 4 in the previous version). In addition, similar to the 3RE and Δ HIRAN variants, the impaired PCNA ubiquitination by these *K/Rad5* variants can be further suppressed by dsDNA or ssDNA, suggesting that the HIRAN domain and additional regions in *K/Rad5* contribute to DNA-based regulation of PCNA ubiquitination.

In summary, the above new data and data presented elsewhere in the manuscript indicate that the positively charged region in the HIRAN domain and the HIRAN-Snf2 interaction both play critical roles in Rad5's PCNA and DNA binding activities and the Rad5-catalyzed PCNA ubiquitination. As such, our work not only reveals different activities of the HIRAN domains in HLTF and Rad5 proteins, but also provided structural and biochemical explanation for these distinct features. We have revised the second last paragraph in the "Discussion" section to include the above discussion.

We updated supplementary figure 11 panels d and h, so that all *K/Rad5* forms in these figures contain Histag that does not change *K/Rad5*'s activities. We moved the section of "The Snf2-HIRAN interaction contributes to Rad5's multiple activities" to the end of "Results", as assays added to this section were based on findings made in other sections in "Results".

My other comments are listed according to my original points and the authors' responses:

1. To ensure the overall stability of the tested mutants it would be helpful to include some QC

data of the purifications (e.g. SEC profiles). In addition, as mentioned above, conclusions about the contribution of the different Rad5 domains on its activities would require a systematically screening of all Rad5 activities with the respective mutants is required (i.e. including interactions with PCNA, ssDNA and dsDNA).

We added a supplementary figure 2 to present SEC profiles of *Kl*Rad5 and its variants. As mentioned above, we have systematically screened the ATPase, ubiquitin ligase, PCNA binding and ubiquitination, ssDNA and dsDNA binding activities of *Kl*Rad5 variants with disrupted HIRAN-Snf2 interaction. In the previous version of the manuscript, we have presented experiments to characterize these activities of the 3RE and Δ HIRAN variants designed to probe the function of the HIRAN domain, except for the ATPase activity. We have characterized the ATPase activity of these variants and found that they are reduced compared to the wild type *Kl*Rad5. This data is presented in the new supplementary figure 8e.

2. Figures S7D and E are still difficult to understand. The authors should revise the labels and figure captions. Especially, they should clearly indicate the associated tags on the respective proteins and indicate which protein was used as the bait in the co-precipitations.

We modified the figure and its legend accordingly. Supplementary figure 7e in the previous version is now supplementary figure 7f.

3. The authors have clarified why they used ScRad5 in the replication fork regression assays. Nevertheless, definitive evidence that would allow conclusions about two subfamilies of Rad5 relatives, it would be required to perform this assay with the KlRad5 protein.

We performed the suggested experiments and found that *K. lactis* Rad5 possesses robust fork regression activity. The activity is suppressed by the Q1051D substitution that inhibits ATP hydrolysis by its Snf2 domain, consistent with previous reports. Similar to ScRad5, fork regression by *K. lactis* Rad5 is also severely inhibited by the 3RE substitution, suggesting that the positively charged region in the HIRAN domain is critical for this reaction. These new data are presented in the new Supplementary figure 10.

4. The results regarding the ubiquitin ligase activity of Rad5 in the presence or absence of ssDNA and dsDNA appear inconsistent. The inhibitory effect of ssDNA and dsDNA is only observable when the enzymatic activity of Rad5 is restricted to the addition of one ubiquitin to Ub-PCNA ("priming" Ub) (Figure S9B). In figure S9A, with non-restricted K63-chain formation activity on Ub-PCNA, no inhibitory effect is detectable. This seems to indicate that the inhibitory effect is only present for the priming Ub on Ub-PCNA. The authors should discuss and try to resolve this. Regarding the effect of PCNA loading: the study quoted by the authors showed that loading has a stimulatory effect on Rad5's ubiquitin ligase activity towards PCNA. I am not suggesting that the authors repeat their assays with loaded Rad5, as this may go beyond the scope of the manuscript, but they should discuss that their assays trying to tease apart ssDNA binding and PCNA binding/ubiquitylation are likely non-physiological, given that the physiological substrate would be loaded PCNA.

Supplementary figure 9a shows that dsDNA and ssDNA have inhibitory effects towards the formation of long PCNA-anchored ubiquitin chains that migrate to the upper portion of the gel. The bands in this region especially near the upper gel edge are clearly weaker for reactions

with dsDNA or with ssDNA but without SSB, indicative of reduced long PCNA-anchored ubiquitin-chain formation. We agree with the reviewer that the inhibition of the Rad5-catalyzed PCNA ubiquitination by DNA we observed may be non-physiological, as PCNA is not loaded to dsDNA in our experiments. We have revised the related paragraph 3 in the “Discussion” section to include this discussion.

5. The authors have responded well to this point.

6. The authors have responded well to this point. Yet, the vastly different propensities of yeast and human cells to undergo fork regression (in yeast these have been directly observed only in checkpoint mutants, reflecting a very non-physiological situation) might suggest that the distinct properties of the two classes of Rad5 relatives in yeast versus humans could be related to that phenomenon.

We appreciate this insightful comment and have revised the first paragraph in the “Discussion” section to include the suggested discussions.

7. Some typos remain:

Abstract: “automatous” should probably be “autonomous”

Page 3: “an universal” should be “a universal”

Page 4: “We present here crystal structure of ...” should be “We present here a crystal structure of ...”

Page 7: “florescent” should be “fluorescent”

We thank the reviewer for pointing out these typos and have corrected them.

8. The authors have responded well to this point.

9. The detection method (Figure 3c) is still not mentioned (Coomassie staining?).

We have revised the legend to indicate the detection method, which is Coomassie staining.

Additional minor comments:

Figure S5d: The authors should include input samples, loading control and indicate the detection method.

We have modified the figure to include the input samples and loading control.

Figure S8a: The authors should describe the rationale of using single-strand binding protein (SSB) in the figure caption or the main text.

We used SSB to eliminate the possibility that the observed *Kl*Rad5-ssDNA co-precipitation is due to the interaction between *Kl*Rad5 and the avidin resin or other components in the binding reaction. We have revised the first paragraph in the “Rad5’s HIRAN domain contributes to DNA binding” section to include the above rationale.

Reviewer #2 (Remarks to the Author):

The manuscript was carefully revised by taking into account of all of my concerns and comments. The revised version provides more in-depth insights into functions of Kl-Rad5 and highlights the functional difference between Rad5 and HLTF. Several concerns are newly arisen in the revised manuscript.

We appreciate the positive comments from this reviewer. We have performed additional experiments and revised the manuscript to address the points he/she raised, as detailed below.

1) Binding experiments revealed that the positive region in HIRAN of Kl-Rad5 interacted with the acidic region of PCNA, where D109 and E113 were crucially involved in the interaction. These residues are located in the cleft between PCNA monomers. Based on the trimer structure of PCNA, these residues might be involved in the trimer assembly. In fact, E113A mutation in ScPCNA reduced trimer stability of ScPCNA (Dieckman et al., DNA Rep., 2013). The author should evaluate the assembly of Kl-PCNA with E113K mutation by SEC. The reviewer is afraid that instability of PCNA trimer might affect the interaction.

We have performed the suggested SEC experiments. The elution profiles of the D109K- and E113K-substituted PCNA are highly similar to the wild type PCNA, suggesting that the substitutions do not alter the overall PCNA structure. These data are presented in the new supplementary figure 7e.

2) The reviewer agrees that ssDNA binding is a common feature for HIRAN domain in Rad5 family members (page 10, line 222-223). In the revised manuscript, the authors reveals that Kl-HIRAN binds to ssDNA and dsDNA with blunt ends or 5'- or 3'- overhanging ssDNA regions. And the affinity to ssDNA and dsDNA with overhanging ssDNA regions are much stronger. In this context, the readers might consider that HIRAN of human HLTF has no affinity for dsDNA with blunt ends or 5'- or 3'- overhanging ssDNA regions. However, these results of Kl-HLTF are consistent with those of HIRAN of human HLTF (Hishiki et al. JBC, 2015). To avoid misleading, the author should add description about affinity of HIRAN of HLTF for dsDNAs.

We have revised the third paragraph in the “Rad5’s HIRAN domain contributes to DNA binding” section to include a description of the mentioned study.

Minor comment:

The labels “ kT/e ” in the electrostatic potential may be “ k_B/e ”, where “ k ” and “ B ” are shown as an italics and a subscript, respectively, because k_B would be Boltzmann constant. The reviewer overlooked that in the first review.

We thank the reviewer for pointing out this error. The Boltzmann constant is usually presented in italic lowercase k or k_B as described by the reviewer. The unit for the electrostatic potential is kT/e , in which T is the absolute temperature and e is the charge possessed by an electron. We have corrected this mistake in the related figures.

Reviewer #3 (Remarks to the Author):

The authors have redone the HDXMS experiments and they now appear trustworthy. The differences in deuteration level with and without dsDNA bound are rather small however, although statistically significant. Bigger differences may be obtained with further experimentation with shorter incubation times, lower incubation temperature or through increased fraction of dsDNA-bound KIRAD5 relative to unbound KIRAD5 during labeling with deuterium.

Morten Beck Trelle

We thank the reviewer for the positive comments and his/her help in improving the HDXMS data.

REVIEWERS' COMMENTS

Reviewer #1 (Remarks to the Author):

The authors have addressed all my remaining concerns to my satisfaction.

Reviewer #2 (Remarks to the Author):

To Authors:

The manuscript was carefully revised by taking into account of all of my concerns and comments. The authors performed SEC analysis of PCNA mutants and the result supported the conclusion of the interaction between KI-HIRAN and KI-PCNA.

Minor comment in the pervious review lacked absolute temperature "T". The authors correctly revised the dimension in the new version. The reviewer only point out typos.

Page 11 in PDF;

Lines 255 and 261, catalzyed => catalyzed

Hiroshi Hashimoto

Reviewer #1 (Remarks to the Author):

The authors have addressed all my remaining concerns to my satisfaction.

Reviewer #2 (Remarks to the Author):

To Authors:

The manuscript was carefully revised by taking into account of all of my concerns and comments. The authors performed SEC analysis of PCNA mutants and the result supported the conclusion of the interaction between KI-HIRAN and KI-PCNA.

Minor comment in the pervious review lacked absolute temperature “T”. The authors correctly revised the dimension in the new version. The reviewer only point out typos.

Page 11 in PDF;

Lines 255 and 261, catalzyed => catalyzed

Hiroshi Hashimoto

Our response, we thank Dr. Hashimoto for pointing out these typos and have corrected them.